

# Continuous high resolution mid-latitude belt simulations for July-August 2013 with WRF

Thomas Schwitalla[1], Hans-Stefan Bauer[1], Volker Wulfmeyer[1], and Kirsten Warrach-Sagi[1]

[1]Institute of Physics and Meteorology, University of Hohenheim, Garbenstrasse 30, 70599 Stuttgart, Germany

*Correspondence to:* Thomas Schwitalla (thomas.schwitalla@uni-hohenheim.de)

**Abstract.** The impact of a convection permitting (CP) northern hemisphere latitude-belt simulation with the Weather Research and Forecasting (WRF) model was investigated during the July and August 2013. For this application, the WRF model together with the NOAH land-surface model (LSM) was applied at two different horizontal resolutions, 0.03° (HIRES) and 0.12° (LOWRES). The set-up as a latitude-belt domain avoids disturbances that originate from the western and eastern boundaries and therefore allows to study the impact of model resolution and physical parameterizations on the results. Both simulations were forced by ECMWF operational analysis data at the northern and southern domain boundaries and the high-resolution Operational Sea Surface Temperature and Sea Ice Analysis (OSTIA) data at the sea surface. The simulations are compared to the operational ECMWF analysis for the representation of large scale features. To compare the simulated precipitation, the operational ECMWF forecast, the CPC MORPHing (CMORPH), and the ENSEMBLES gridded observation precipitation data set (E-OBS) were used.

Compared to the operational high-resolution ECMWF analysis, both simulations are able to capture the large scale circulation pattern though the strength of the Pacific high is considerably overestimated in the LOWRES simulation. Major differences between ECMWF and WRF occur during July 2013 when the lower resolution simulation shows a significant negative bias over the North Atlantic which is not observed in the CP simulation. The analysis indicates deficiencies in the applied combinations of cloud microphysics and convection parametrization on the coarser grid scale in subpolar regions. The overall representation of the 500 hPa geopotential height surface is also improved by the CP simulation compared to the LOWRES simulation apart across Newfoundland where the geopotential height is higher than in the LOWRES simulation due to a northward shift of the location of the Atlantic high pressure system.

Both simulations show higher wind speeds in the boundary layer by about 1.5 m s$^{-1}$ compared to the the ECMWF analysis. Due to the higher surface evaporation, this results in a moist bias of 0.5 g kg$^{-1}$ at 925 hPa in the planetary boundary layer compared to the ECMWF analysis. Major differences between ECMWF and WRF occur in the simulation of the 2-m temperatures over the Asian desert and steppe regions. They are significantly higher in WRF by about 5 K both during day- and night-time presumably as a result of different soil hydraulic parameters used in the NOAH land surface model for steppe regions.

The precipitation of the HIRES simulation shows a better spatial agreement with CMORPH especially over mountainous terrain. The overall bias reduces from 80 mm at the coarser resolution to 50 mm in the HIRES simulation and the root mean square error is reduced by about 35% when compared to the CMORPH precipitation analysis. The precipitation distribution agrees much better with the CMORPH data than the LOWRES simulation which tends to overestimate precipitation, mainly



caused by the convection parametrization. Especially over Europe the CP resolution reduces the precipitation bias by about 30% to 20 mm as a result of a better terrain representation and due to the avoidance of the convection parameterization.

## 1 Introduction

On longer time scales like seasonal, decadal, and climate predictions, global General Circulation Models (GCM) are commonly applied with a typical horizontal resolution in the range of 1–2°(Taylor et al., 2012, e.g.). Since it is often desired to have higher resolutions over a region of interest to better represent the land-surface interaction, more and more regional climate models (RCMs) covering only a subregion of the globe are still applied at a resolution between 0.1° and 0.5°.

In the Coordinated Regional Downscaling Experiment CORDEX (http://www.cordex.org; Giorgi et al., 2009), several RCMs are applied with grid distances of 0.44° for different continental scale regions around the globe at affordable computing power. As this resolution may still suffer from a too coarse horizontal resolution, e.g., the EURO-CORDEX project (http://www.euro-cordex.net/) focuses on regional climate simulations for Europe at 0.11° resolution. Studies of Kotlarski et al. (2014) and Vautard et al. (2013) indicated that increasing the resolution from 0.44° to 0.11° results in beneficial or detrimental effects with respect the simulation of 2-m temperatures with biases in the range of ±2 K. However, Kotlarski et al. (2014) show a large model variability with respect to convective precipitation during the summer season over Europe.

Heikkilä et al. (2011) applied the Weather Research and Forecasting model (WRF) over Norway at 0.33° and 0.11° showing a superior performance of the 0.11° domain with respect to precipitation and 2-m temperatures. Warrach-Sagi et al. (2013) performed a 20 year simulation with the WRF model over Europe at 0.33° and 0.11° resolution where the focus was set on precipitation in Germany. Their study shows an overestimation of precipitation and a higher wet day frequency than observed. The 0.11° simulation shows the windward-lee effect in the low mountain ranges in Germany also observed in a study of Schwitalla et al. (2008) who performed simulations at 7 km horizontal resolution using the MM5 model Grell et al. (1995). Due to the application of a convection parameterization, convection was triggered too early with underestimated peak precipitation rates.

As 0.11° resolution can still be too coarse to resolve orographic precipitation, Warrach-Sagi et al. (2013) applied the WRF model with a resolution of 0.0367° during the Convective and Orographically-induced Precipitation Study (COPS;Wulfmeyer et al., 2011) period in Summer 2007. Their study demonstrated a significant improvement with respect to the spatial distribution of precipitation when applying a convection permitting (CP) resolution due to the better resolved terrain and explicit treatment of deep convection. A better spatial distribution of precipitation was also observed in studies of Bauer et al. (2011), Prein et al. (2013), Warrach-Sagi et al. (2013), and Piere et al. (2015) who clearly identified the benefit of performing convection permitting simulations.

RCMs are either driven by coarser scale models, GCMs, or coarser scale reanalysis data like ERA-Interim (Dee et al., 2011). Therefore the numerical solution of a RCM is driven by the lateral boundary conditions (LBCs) given by the driving model. As the inflow boundaries at coarser grid scales may be imperfect, this can deteriorate the results of the RCM. Laprise et al. (2008) suggested that RCMs require a large model domain to capture all the fine scale features especially in the upper tro-





posphere in mid-latitudes. Schwitalla et al. (2011) evaluated the performance of a limited area WRF set up on a large scale driven precipitation event in summer 2007. The WRF model showed superior performance with respect to the representation of precipitation compared to the smaller domain operational numerical weather prediction (NWP) model of the German Meteorological Service.

Diaconescu and Laprise (2013) tested the effect of different domain sizes on large scale features with simulations at $\sim 0.5°$ horizontal resolution. When RCMs are driven by LBCs containing errors, RCMs can reduce errors in the large scale circulation by applying a large model domain. Problems still can occur as the driving models often contain different physics schemes than the LAM leading to inconsistencies at the boundaries which can penetrate into the model domain.

An option to partially overcome the necessity to apply boundary conditions from a coarser LAM or GCM are channel or

latitude-belt simulations. With this type of simulations, it is only required to apply LBCs on the northern and southern boundaries. A typical application is a tropical channel covering an area between 30° S and 30° N. Due to computational constraints, these simulation often have a resolution between 20–30 km (e.g. Coppala et al., 2012; Evan et al., 2013; Fonseca et al., 2015). One idea of this special type of simulations is to allow storm systems to cross a whole ocean basin without being truncated by domain boundaries. As the general circulation is west-east oriented, e.g. errors in the large scale circulation patterns can be

traced back to the applied physics schemes.

Europe is frequently affected by storm systems transiting from Newfoundland towards Europe. By applying RCMs, western LBCs can destroy certain features of these storms before they reach Ireland and the Western Europe. Žagar et al. (2013) performed one of the first higher resolution latitude belt simulation covering the northern hemisphere between 35° N and 70° N. They applied the WRF model for a three month period covering January-March 2009. The horizontal resolution was

0.25° which, at this time, was very close to the horizontal resolution of the ECMWF operational model used to force the lateral boundaries. To show the benefit of such a latitude belt, they performed additional LAM simulations with different west-east stretching domain sizes. Their results show the largest uncertainties over the Atlantic and Pacific due to an imperfect nesting of WRF, which means that the model physics between WRF and the driving model differ.

The goal of this study is to evaluate the added value of a convection permitting simulation without any deterioration by

LBCs in west-east direction. So far no forecasts on the convection permitting scale in a latitude belt configuration have been performed.

We are investigating whether a very high-resolution latitude belt domain improves the long-term skill with respect to the large-scale circulation and especially precipitation. In our work, we are addressing the following questions:

– What is the benefit of a CP resolution with respect to the spatial representation of large scale features in comparison to

coarse resolution?

– Does the higher resolution lead to an improvement of surface variables such as 10-m wind speed and 2-m temperatures?

– What is the benefit of the CP resolution regarding the spatial distribution and amount of precipitation?

This study organized as follows: Section 2 gives an overview about the technical details and the experimental set-up followed by a review of the weather situation during the simulation period in section 3. In section 4, a comparison of the large



scale circulation against ECMWF operational analysis followed by a comparison of 2-temperatures, 10-m wind speeds and precipitation will be performed. Section 5 provides a discussion of the results. The final section 6 summarizes our results.

## 2 Experimental setup

### Model setup

For the experiment, the limited area WRF model (Skamarock et al., 2008) version 3.6.1 was applied. Due to the greater variety of physics options, the fully compressible non-hydrostatic Advanced Research WRF (ARW) is used in this study. In contrast to the most commonly applied limited area grids, a latitude belt was selected for this study. This latitude belt covers the northern hemisphere between 20° N and 65° N and is shown in Figure 1. This is the typical latitude range for weather systems affecting Europe.

Two configurations with a latitude-longitude grid are selected: a simulation with 0.12° resolution where convection was parametrized (hereafter named LOWRES) and a convection permitting configuration consisting of 12000×1500 grid cells with a horizontal resolution of 0.03° (hereafter named HIRES). The reason to choose a 0.12° resolution is that the current resolution of the ECMWF operational model is similar and it is also similar to the resolution applied in the EURO-CORDEX experiment (Giorgi et al., 2009, e.g.).

Both simulations were performed as dynamical downscaling of the ECMWF analysis with 57 vertical levels, of which 14 levels were within the first 1500 m above ground level, and the model top was set to 10 hPa. The numerical time step was 10 s in the HIRES simulation and 40 s in the LOWRES simulation in order to avoid Courant-Friedrichs-Lewy (CFL) criteria violations in the northern part of the model domain. In addition, the *epssm* parameter ($\beta$ in the study of Dudhia, 1995) was set to 0.5. This parameter biases the average in vertical wind speed for sound wave computation leading to an increased stability

when the terrain slope is steep.

WRF-ARW offers multiple physics parametrizations. The surface layer above the ground is parametrized by the revised MM5 surface layer scheme of Jimenéz et al. (2012) and is combined with the YSU boundary layer scheme of Hong (2007). The YSU is widely used and extensively evaluated in the WRF community (Nolan et al., 2009; Schwitalla et al., 2011; Shin and Hong, 2011; Milovac et al., 2016, e.g.).

Cloud microphysics are parametrized by the Morrison two-moment scheme (Morrison et al., 2009) which includes prognostic variables for liquid and frozen hydrometeors and their corresponding number concentrations. The Morrison scheme is a full 2-moment scheme which is beneficial to represent summertime convection where frozen particles can collect liquid water. This scheme was used during summertime convective precipitation events as shown in studies by Schwitalla and Wulfmeyer (2014) and Bauer et al. (2015a).

For the 0.12° simulation, the Kain-Fritsch (KF) cumulus scheme (Kain, 2004) together with the default trigger function was applied. The RRTMG longwave and shortwave schemes of Iacono et al. (2008) were applied to parametrize radiation transport. In addition to cloud water, cloud ice and snow, RRTMG interacts with rain water. Shallow convection was parametrized by the GRIMS scheme of Hong et al. (2013) in both simulations. At the lower boundary, the WRF model is coupled to the land



surface model (LSM) NOAH (Chen and Dudhia, 2001; Ek et al., 2003). The different physics options are summarized in Table 1.

The representation of the soil texture is crucial when performing simulations on higher resolution. Studies of Warrach-Sagi et al. (2008, 2013), and Acs et al. (2010) indicated that the global FAO (Food and Agricultural Organization of UNO) soil texture
data set, which has a resolution of 5' (approx. 10 km), shows significant deviations from high-resolution soil databases. When approaching the convection permitting scale, a soil texture data set at the corresponding resolution is required since the texture determines the soil moisture. We used a modified soil texture data set from Milovac et al. (2014) which is derived from the Harmonized World Soil Database (HWSD) between 60° N and 60° S available at 1 km resolution. Land cover is described by the 20-category MODIS data set from the International Geosphere Biosphere Programme (IGBP) program available at
30″ resolution.

The simulations were performed for a 2 month period starting at 01 July 2013 00 UTC. Forcing data at the northern and southern boundaries were provided by 6h ECMWF operational analysis data on model levels. The LBCs are blended with the default linear decay over five grid points into the WRF model grid. Sea Surface temperatures were provided by the high-resolution Operational Sea Surface Temperature and Sea Ice Analysis (OSTIA) data Donlon et al. (2012).
Soil moisture and temperature were initialized from the ECMWF operational analysis. The Hydrology land-surface model HTESSEL (Balsamo et al., 2009) of ECMWF operational system includes seven different soil textures which are in a better accordance with the soil textures used in the NOAH LSM as compared to e.g. the study of Žagar et al. (2013) where the old TESSEL (Viterbo et al., 1999) was available at ECMWF. Previous studies showed that NOAH's soil moisture and temperatures spin up within 10–14 days in Europe when initialized with HTESSEL. A soil moisture spin-up run was not performed as the
main focus of this study is to compare the different resolutions on a rather short time scale of two months.

**Computational aspects**

The WRF model simulations were performed at the High Performance Computing Center Stuttgart (HLRS) on the Cray XC40 (http://www.hlrs.de/systems/cray-xc40-hazel-hen/). At the time when the simulations were performed, the system consisted of approx. 4000 compute nodes each equipped with 2 Intel 12-Core CPUs with 2.5 GHz clock frequency. The model was
compiled with version 14.7 of the Portland Group compiler, Cray MPI 4.3.2 and parallel NetCDF 1.5.0. The total number of cores was partitioned in such a way that each node was filled with four MPI tasks and six OpenMP threads so that in total 14000 MPI tasks were used.

The Lustre file system was configured so that 128 object storage targets (OSTs) were used for writing into a single NetCDF file of 92GB size for the HIRES simulation. Further testing revealed that it was not beneficial to use more MPI tasks as this
deteriorated the I/O rate which was in the range of $\sim$ 6–7 GByte/s. The total data amount including restart files for the HIRES simulation is about 300TB.

The necessary input fields from the ECMWF analysis are about 454 GB in size. It has to be noted that due to limitations in the WPS code of the WRF model system, each GRIB2 file has to be smaller than 2 GB as otherwise this file cannot be fully read in by the ungrib program. The required high-resolution SST data are about 10 GB in total. As they are only available in





daily intervals, these data have to be interpolated to the 6 h intervals of the ECMWF analysis. This interpolation was performed by using version 1.7.0 of the Climate Data Operators (CDO;https://code.zmaw.de/projects/cdo). The time required to download and postprocess these data is about 3 days.

As the Metgrid interpolation program requires a lot of memory and does not support Parallel NetCDF, the input fields had to
be splitted using a value of 102 for `io_form_metgrid` in *namelist.wps* so that each MPI task writes its own small NetCDF file. These files were ingested into the Real program using `io_form_input=102` in the *namelist.input* and the required wall time for Real was about 24 h. The used *namelist.input* for the HIRES simulation is shown in the Appendix.

For the HIRES simulation, 3500 compute nodes (84000 Cores) were used for 3.5 days wall time resulting in ∼0.15 s for each model time step giving a speed up of 66 compared to real time. The 0.12° simulation was performed on 120 compute
nodes and was finished within 31 hours wall time.

If such a high-resolution simulation is considered for operational applications, users have to reduce the output frequency considerably as otherwise the time for writing the files becomes the prevailing process. If even a higher number of grid points is planned to use, one has to take care about the NetCDF limitations in the commonly used CDF-2 format. This convention only allows $2^{32}$-4 bytes per array which can be too small for future experiments so that the new CDF-5 standard has to be
considered. Another alternative is to use NetCDF4 with HDF5 support but due to the applied compression, this may require the same time for writing although the file sizes may be considerably smaller than with classic NetCDF. Further information about technical challenges can be found in (Bauer et al., 2015b, e.g.).

## 3   Seasonal statistics

In order to classify the meteorological conditions of summer 2013, climatologies from the ERA-Interim analysis (Dee et al.,
2011) were analyzed. Figure 2 displays the mean 500 hPa geopotential together with the anomaly of July and August 2013 compared to 1979–2012 (contour lines). It shows the subtropical high with a geopotential height of more than 5900 gpm over the Central Atlantic which is in accordance with the climatological mean of 1979–2012.

Over the northern mid-latitudes, a positive anomaly of the 500 hPa geopotential height is observed over Newfoundland, western United States and the northern Pacific while especially over Europe the 500 hPa geopotential height is significantly
higher than the climatological mean. In connection with lower geopotential values over Greenland this leads to stronger wind speeds in the mid-troposphere and thus changes the circulation pattern compared to the climatology. The stronger gradient in the 500 hPa geopotential between the northern Pacific and East Asia support the transport of warm and moist air masses towards North and East Asia, especially as the SSTs are higher than the climatological average during both months (Fig. 3). The location of the jet stream is similar during the simulation period as compared to the climatology. The most remarkable
difference is the considerably increased wind speed east of Newfoundland and over Central Asia while the wind speeds are weaker over the Pacific (not shown)

The mid-troposphere Azores High extends further towards Central Europe and is also visible at the surface. Fig. 4 shows the average MSLP (contour lines) for the two month period together with the corresponding anomalies (shaded). The MSLP





anomaly reaches 3 hPa over Central Europe while at the same time the MSLP bias is negative between Greenland and Iceland leading to higher low level wind speeds than normal. July and August 2013 were characterized by a strong positive Northern Oscillation (NAO) index of 0.7 and 1.0 in July and August, respectively. Apparently, the positive SST anomalies (Fig. 3) around 2.5°C over the the Central Atlantic and the northern Pacific are responsible for the lower MSLP over the North Atlantic and the

Eastern Pacific as compared to the climatology. Particularly in Europe, there was a heat wave in July leading to dry conditions in Western Europe (Dong et al., 2014). The precipitation amounts over Central and Northern Europe were less than 50% of the climatological mean as indicated by the E-OBS (Haylock et al., 2008) precipitation data set (not shown).

## 4  Results

Obtaining consistent observations of wind, temperature, and humidity at different altitudes, which are on a comparable reso-
lution to 0.12°, is currently very challenging. They are only available for a few countries and are not homogeneous. Satellite derived products like integrated water vapor, radiation data, and cloud products are often available on grids with resolutions coarser than 0.5°. Also they are mostly only available for a certain region depending on the satellite coverage. Therefore we decided to use the operational ECMWF analysis to compare the results of both WRF simulations with respect to the large scale patterns apart from precipitation where suitable data sets are available.

The ECMWF analysis is generated by a four-dimensional variational data assimilation system (4DVAR; Bouttier and Courtier, 1999). It combines a model background field from a previous forecast with high-resolution observations in order to obtain a high quality gridded analysis field. The 4DVAR at ECMWF includes several different observation types like surface measurements, radio soundings, satellite radiances and aircraft measurements. ECMWF reanalysis data (Dee et al., 2011), available on a 0.75° grid are widely used for verification of RCMs (Vautard et al., 2013; Warrach-Sagi et al., 2013;
Katragkou et al., 2015, e.g.) as they can easily be obtained. As this resolution is too coarse for this high-resolution simulations, we decided to use the operational ECMWF analysis for comparison. The regridding of the WRF output to the ECMWF grid at 0.125° for was performed with an MPI-compiled version of the Earth System Modelling Framework (ESMF) RegridWeightGen tool[1] within the NCAR Command Line (NCL) framework.

For the verification of precipitation, the CPC MORPHing technique (CMORPH;Joyce et al., 2004) data set was applied. It is
an almost global precipitation analysis, based on low orbit microwave satellite data. In version 1.0, this product is bias corrected and also uses surface precipitation data where available (blended product). The daily precipitation analysis is available on a 0.25°×0.25° grid. Studies of e.g. Liu et al. (2015) and Stampoulis et al. (2013) show a reasonable correlation of the CMORPH precipitation analysis with ground stations in different regions around the globe. In order to compare the observations with the simulations, the HIRES and LOWRES data were interpolated to the CMORPH grid. For Europe, the E-OBS precipitation data
set (Haylock et al., 2008) was selected. A recent study of Skok et al. (2016) showed a superior performance of the E-OBS data over Europe compared to the CMOPRH observations.

---

[1]https://www.earthsystemcog.org/projects/regridweightgen/





As both WRF simulations are performed without data assimilation, it cannot be expected that they represent single extreme weather events. Nevertheless both simulations are expected reproduce to the large scale weather situation reasonably well with advantages when applying a convection permitting resolution.

### 4.1 Large scale circulation

Figure 5 shows the comparison of the averaged MSLP at 12 UTC for July and August 2013. In July the ECMWF model shows a strong high pressure system over the Eastern Pacific and a well defined high pressure system over the Atlantic. This is a typical situation during the summer over the northern hemisphere (see Fig. 4).

During July, both WRF simulations are able to capture the general features compared to the ECMWF analysis. Larger differences occur over the Pacific Ocean and over the Central Atlantic where the high pressure systems are located. The

intensity of the Pacific high is significantly overestimated in both simulations (Fig. 5g,i) and its location is slightly shifted to the south showing a dipole structure. This behavior was also observed in a study of Cassano et al. (2011) who performed month long simulations using WRF over the polar and subpolar region.

In both simulations the Atlantic high pressure system extends further to the north towards the Azores islands and also the intensity is overestimated as compared to the ECMWF analysis. The LOWRES simulation shows a low pressure area east of

Greenland. This is not simulated in the HIRES simulation which is in a better accordance with the ECMWF analysis.

During August 2013 (right column of Fig. 5) the Pacific high is still overestimated with a bias of more than 5 hPa (Fig. 5h,j) . The high pressure system over the Atlantic shows a different shape compared to the ECMWF analysis (Fig. 5b,d,f). The strong negative bias over Central Asia is the result of too high 2-m temperatures (see later in section 4.3).

Both WRF simulations show different sensitive regions compared to ECMWF as indicated by the MSLP standard deviation

shown in Figure 6 for the different months. The LOWRES experiment exhibits an unrealistically large variability over the Hudson Bay in July associated with a stronger variability of the 850 hPa wind speeds (not shown). The large standard deviation over the North Atlantic shown by the HIRES simulation can be explained by a higher internal variability due to the higher resolution. Such variability cannot be expected in the ECMWF analysis due to its coarser resolution which does not resolve the belonging high-resolution dynamical processes. Nevertheless the overall location of the high standard deviation areas in

the HIRES simulation is in good agreement with the ECMWF analysis.

The variability of the MSLP over the Western Pacific is significantly overestimated by both WRF experiments. The LOWRES experiment seems to have an even higher tendency to develop tropical storms as seen by the simulated corridor of higher standard deviation. The very high standard deviation over the Aleutian Islands may be related to the higher resolution as these islands consist of volcanoes with elevations up to 2000 m. As they are only partially resolved in the ECMWF model

and especially its 4DVAR system running at even coarser resolution, this can explain the different behavior in this region. In combination with higher resolution SSTs and a better represented landmask this also contributes to higher sensitivities.

Especially in August, the LOWRES simulation tends to exaggerate the development of tropical storms as indicated by the large standard deviation of more than 12 hPa south of Japan. Apparently this is related to the stronger pressure gradient over



the West Pacific (Fig. 5). According to the analysis of the Japanese Meteorological Agency (JMA), only two tropical storms were present during August in the West Pacific north of 20° N.

Figure 7 shows the mean 300 hPa wind speed of both WRF simulations compared with the ECMWF analysis for 12 UTC. During both months, the ECMWF analysis (Fig. 7) shows a well defined subtropical jet stream north of the Tibetan Plateau

with an average wind speed of 30 m s$^{-1}$ over Central Asia which is typical for the monsoon season. Also the polar jet over the Pacific and Newfoundland is clearly visible. Compared to the climatology from 1979–2012, the position of the subtropical and polar jet is very similar, with considerably higher wind speeds along the subtropical jet.

The large scale structure is captured in both WRF simulations while the HIRES simulation shows a weaker maximum over Central Asia as compared to ECMWF. They tend to overestimate the intensity of the subtropical jet over the North Pacific but

the HIRES simulation has a lower RMSE as compared to the LOWRES experiment. At the 200 hPa level (not shown here), the wind maximum over Central Asia is simulated more accurately in the HIRES simulation. This indicates a possible influence of the better resolved terrain over Asia. As the surface low over the Tibetan plateau is deeper than observed, this can induce a force which moves the subtropical jet further to the north deforming the subtropical jet as shown by the reddish colors over the northwest of China and Mongolia in Fig. 7. In addition, the better representation of the Pamir and Tien Shan Mountains in the

higher resolution model also play a role in terms of blocking the backward motion of the jet.

To complement the results for the large scale circulation, Figure 8 shows the mean 500 hPa geopotential height of the ECMWF analysis at 12 UTC time steps. Here a wave like structure with 5–6 stationary waves is visible during both months in the analysis indicated by the alternating reddish and red colors.

In July, the general features agree in both WRF simulations and the differences partially reflect the the displacement of

the low pressure systems shown in Fig. 5. The LOWRES simulation simulates high geopotential over Mongolia and the West Pacific in July 2013 as compared to the ECMWF analysis. This bias further increases in August 2013 exceeding 100 gpm over the North Pacific and Newfoundland (Fig. 8d) as a result of the even stronger displacement of the pressure systems similar to the results of Cassano et al. (2011). The HIRES simulation also simulates high geopotential at 500 hPa over the Atlantic but the differences over the West Pacific remains much smaller in August 2013 compared to the LOWRES simulation.

In addition, Figure 9 shows the time series of the averaged MSLP over the North Atlantic between 40° N and 65° N and 60° W and 10° E (white rectangle named Atlantic in Fig. 1). During the first ∼ 10 days, the HIRES simulation (red line) agrees well with the ECMWF analysis while the LOWRES simulation show slightly lower pressure values. After this period, the LOWRES simulation shows considerably lower MSLP compared to the ECMWF analysis while the HIRES simulation is much closer the ECMWF analysis until day 18 of the forecast where both simulations miss the development of a depression.

Both simulations are able to capture the pressure drop after 25 days of forecast but the HIRES simulation shows a better agreement with the ECMWF analysis. In the further course, both WRF simulation overestimate the strength of the high-pressure situation with being closer to the analysis again after 45 days. Overall, the LOWRES simulation shows a tendency to even further overestimate the strength of low and high pressure systems.



## 4.2 Temperature and moisture in the lower troposphere

The moisture availability in the boundary layer is an important factor for the development of convection and precipitation. As an example, Figure 10a shows the mean 925 hPa water vapor mixing ratio of the ECMWF analysis at 12 UTC. The areas with high moisture availability over India during the monsoon season and the low amount of water vapor over continental Africa and the African west coast can be recognized by the greenish and blueish colors.

From Figure 10 it is seen that WRF estimates a higher moisture content over the central Pacific with a strong bias of $\sim 1.5$ g kg$^{-1}$. The same holds for the Gulf of Mexico and the Western Atlantic. There are only minor differences in the moisture content at 925 hPa north of 45° N due small differences in the MSLP field. Both simulations show similar RMSE values with an improvement of about 5% in the HIRES simulation (not shown). The highest deviations from the ECMWF analysis occur at the east coast of Canada, the North Atlantic, and the West Pacific which are the regions with the highest pressure deviations. The higher moisture values at the American east coast can be related to the transportation of humidity from the Gulf of Mexico due to the more intense high pressure system over the Atlantic leading to a stronger southwesterly flow in the lower troposphere. This behavior is sustained at 850 hPa where a similar pattern is observed.

The upper panel of Figure 11 shows the mean temperature of the ECMWF analysis at 925 hPa. The warm air masses transported from the desert towards the Atlantic due to Passat winds can be identified. The LOWRES simulation (Fig. 11b) shows a very strong positive temperature bias exceeding 3 K over Europe, North Africa and the Northwest Pacific. The HIRES simulation also shows a positive temperature bias but it is less pronounced as in the LOWRES simulation and the bias over the Northwest pacific is significantly reduced by about 2 K.

The temperature bias over Newfoundland is caused by the the inaccurate position of the Atlantic high pressure systems which extends too far to the west (see Fig. 5). Due to the different wind direction, warmer air masses from the Gulf of Mexico are advected towards Canada. Another interesting feature is the strong overestimation of 925 hPa temperatures in both WRF simulations at the west coast of California. This is due to an overestimation of wind speeds associated with the stronger pressure gradient which dries out the air coming from Cascade Mountain range.

The temperature bias is even higher in the LOWRES simulation (Fig. 11b) because of even stronger winds in the boundary layer. In general, the LOWRES simulation shows an even higher temperature bias exceeding 5 K over the North Pacific. Note that the average RMSE in the HIRES simulation over Europe is very small with around 3 K at 925 hpa and 850 hPa. Further the LOWRES simulation does not simulate the tongue of cold air extending from the central North Atlantic towards the west of the Canary islands (indicated by the warm bias south of the Azores in Fig. 11b). The LOWRES simulation tends to overestimate the boundary layer wind speeds in combination with the spatial shift of the high pressure system. Due to the strong high pressure and the resulting subsidence, warm air masses are transported from the African desert towards the Canary Islands. In addition the insufficient representation of the terrain in the LOWRES simulation as e.g. the mountains are not represented by the LOWRES simulation. This leads to a different circulation pattern than in the HIRES simulation.





### 4.3 Surface fields

Figure 12 shows the mean 2-m temperature for the 12 UTC time steps. The LOWRES simulation shows hardly any bias over the western half of the model domain during July 2013 while the bias considerably grows in August 2013. The HIRES simulation (lower row of Fig.12) shows hardly any bias over the ocean except over the Mediterranean where the model exhibits a cold bias of ∼1–2 K. In August, both WRF simulations show a similar temperature bias as the pressure gradient at the east coast is very similar.

The 2-m temperatures over the Tarim basin north of the Tibetan Plateau are significantly overestimated as shown in Figs. 12c-f. The simulated skin temperatures (TSK) of both WRF simulations are ∼ 6 K higher than in the ECMWF analysis. As the 2-m temperatures are calculated based on the TSK and the second lowest model level, this leads to higher values. In addition, the warm bias over Africa during daytime turns into a cold bias during night time (not shown).

The 10-m wind speeds show a weak bias over the continents (Fig. 13) while larger deviations occur over the Ocean. Especially in the West Pacific, the LOWRES simulation shows a large bias of about 5 m s$^{-1}$ during both months while the HIRES simulation is closer to the ECMWF analysis. The deviations in the Atlantic are the results of the slightly larger extent of the high pressure system (see Fig. 5).

### 4.4 Precipitation

The upper panel of Figure 14 shows the accumulated precipitation for the 2 month period over land only. Precipitation in most regions is between 50 mm and 300 mm for the 2 months. The precipitation peaks in the summer monsoon dominated Southeast Asia and India. The precipitation in the Southern United States is dominated by moist air mass inflow from the Gulf of Mexico in August 2013 (see Fig. 10).

Overall, precipitation amounts are overestimated in both WRF simulations (Fig. 14b,c) apart from the west coast of the United States.

The LOWRES simulation (Fig. 14b) shows an even stronger overestimation of precipitation in this regions related to the required convection parametrization which is responsible for over 90% of the total precipitation. Also the LOWRES simulation shows a tendency to simulate more widespread precipitation of lower intensities Schwitalla et al. (2008).

The precipitation maximum over the Korean peninsula caused by the East Asian Monsoon and the maximum over Mexico due to the North American Monsoon are also well captured in the HIRES experiment. For the whole model domain, the mean estimated precipitation from CMORPH during the two month period is 137 mm, the HIRES experiment simulates 186 mm, and the LOWRES experiment predicts 219 mm within the two months. The variance in both simulations is notably higher as given by the CMORPH analysis (161 mm) and the RMSE is 188 mm for the HIRES simulation and 207 mm for the LOWRES experiment. The average precipitation amount from the operational ECMWF forecasting system (Fig. 14d) is similar to the LOWRES simulation, although the average of 186 mm is more closer to the HIRES simulation.

Although both WRF simulations show a positive precipitation bias, it is seen from Fig. 15 that the shape of the precipitation distribution is better represented in the HIRES simulation. Especially the secondary peak in the precipitation amounts of



100 mm is not visible in the LOWRES simulation. The positive benefit of the high-resolution is also seen in the scatter plot displayed in Fig. 16. The regression line is showing a systematic bias of ∼ 70–80 mm in both simulations, however the LOWRES regression line has a different slope pointing to an increasing bias with increasing precipitation intensities.

As also Central Europe together with the Alpine region and the Spanish dry region is of interest in terms of natural disasters
caused by droughts and heavy precipitation (Gobiet et al., 2014, e.g.), Fig. 17 displays the accumulated precipitation over Europe for the 2 month period. The E-OBS analysis shows high precipitation amounts induced by orography over southwestern Norway, Central United Kingdom and the Alps with values higher than 175 mm. The low precipitation amount over the Iberian Peninsula with values lower than 20 mm is clearly visible and well simulated by the HIRES experiment. Compared to the mean precipitation of 87.7 mm, the LOWRES simulation overestimates the total precipitation over Europe by 55% while the
HIRES simulation only shows an overestimation of 25% in this region. Especially the low precipitation amounts over Spain and Sweden are much better represented compared to the LOWRES simulation (Fig. 17b).

The precipitation over the Alps is considerably overestimated by almost 100% and also the precipitation amounts over Spain are too high due to the application of a cumulus scheme. In addition, the overestimation due to an inaccurate representation of the terrain is clearly visible in the United Kingdom and southern Scandinavia. Compared to the observation and the HIRES
simulation, the LOWRES experiment does not simulate the rain shadow area over Sweden. Although both WRF simulations show a positive bias, the precipitation distribution is much better represented by the HIRES simulation (Fig. 18).

Summarizing the statistical results, Fig. 19 shows a Taylor diagram for the spatial distribution of precipitation. The different verification regions are marked by the white rectangles shown in Fig. 1. This Taylor diagram combines information about the spatial correlation (azimuth angle) with the normalized centered root mean square error (RMSE, blue circles) and normalized
standard deviation (dashed black circles). A perfect model would be at the point marked REF.

On the global scale, over Europe, and East Asia an indication for applying a CP resolution is given by the lower RMSE and standard deviations. Over Central Asia and North America the benefit is not as clear as the correlation of the LOWRES simulation is better and the bias is not reduced by the higher resolution. Over Africa, the correlation is almost similar but the HIRES simulation tends to slightly underestimate the amount of precipitation.

## 5 Discussion

Especially over the northern Pacific, the lower resolution simulation shows major deficiencies with an overstrong subtropical Pacific high showing a MSLP bias of more than 5 hPa. At coarser model resolutions, this problem was also observed in a study of Cassano et al. (2011) and is significantly reduced when a higher horizontal resolution is applied. Studies of Pai Mazumder et al. (2012) over Siberia and Efstathiou et al. (2013) over Greece observed a tendency to overestimate the in-
tensity of high and low pressure systems when applying the YSU PBL scheme. The different location of the polar jet over the North Atlantic can result in the transport of warmer and moist air masses from the Central Atlantic towards the north. This can enhance convection over the Atlantic having a strong influence on the simulation of precipitation over Europe. It also





intensifies the cyclogenesis which can lead to more severe storms over Europe and also more precipitation over the eastern part of the United States.

As the LOWRES simulation shows a totally different large scale pattern in July 2013 compared to the HIRES simulation, the question arises whether the strong negative pressure bias over the North Atlantic simulation is caused by the combination

of the applied physics scheme at this particular resolution.

A study of Kotlarski et al. (2014) revealed a similar result when comparing the large scale circulation during the summer months averaged over a 20 year period with the same physics combination as for the LOWRES simulation. If a different microphysics scheme as e.g. the WSM6 (Hong and Lim, 2006) is applied, the strong sea level pressure bias is not present anymore as seen in the CRP-GL configuration in Kotlarski et al. (2014). As the strong negative pressure bias is also not visible

in the HIRES simulation. This points towards an unfavorable combination of the Kain-Fritsch convection parameterization with the Morrison microphysics scheme at this particular resolution over the subpolar regions.

Cassano et al. (2011) performed a simulation with exchanging the default Goddard microphysics scheme (Tao and Simpson, 1993) by the Morrison 2-moment scheme combined with the Grell-Devenji convection parameterization (Grell and Dévényi, 2002). The precipitation bias over the Arctic is increased by about 50% by applying this physics combination.

The Morrison scheme uses a fixed cloud droplet concentration of 250 cm$^{-3}$. This concentration is adjusted at every model time step and is set to this constant value at the end of a vertical loop. The ice nucleation follows a formula of Rasmussen et al. (2002) which is primarily designed for mid-latitudes. WRF offers another switch based on observations from the Arctic but this is an on-off switch. Especially the fixed particle concentration can lead to a more intense formation of optically thick clouds reducing the solar irradiation. E.g. in the Polar-WRF model (Bromwich et al., 2013) the cloud droplet concentration

is reduced to 50 cm$^{-3}$ to produce fewer liquid water droplets. This points out that it may be necessary to either adjust these parameters according to the latitude when performing simulations in sub-polar regions or to apply a different combination of cloud microphysics and convection parametrizations at coarser resolutions.

The general appearance of the boundary layer humidity fields is comparable with the ECMWF analysis. Both simulations show a positive water vapor bias of 0.6 g kg$^{-1}$ for the HIRES simulation and 0.4 g kg$^{-1}$ for the LOWRES simulation and a

strong moisture bias of more than 2 g kg$^{-1}$ over Newfoundland as this is a sensitive region with respect to the development of low pressure systems. Although the overall bias seems to be fairly small, this can have a meaningful influence on the initiation of convection.

The 925 hPa and 2-m temperatures exhibit a large positive bias in both simulations whereas the bias of the LOWRES simulation is even higher especially over the ocean. The large deviation over the Taclamacan and Gobi desert is the result of

higher surface temperatures. Compared to the analyzed surface temperatures from ECMWF, the simulated surface temperatures are significantly higher in the WRF simulations. A possible explanation could be the different albedo in the WRF simulation and the ECMWF model. The albedo in the WRF model is 0.04 smaller than in the ECMWF analysis thus allowing a 4% higher absorption of radiation and thus higher temperatures (Branch et al., 2014, e.g.). Further, as pointed out by Zhang et al. (2014), the soil hydraulic parameters used in the NOAH LSM show some deficiencies in desert and steppe regions. As our study



already makes use of an improved version of the thermal roughness length calculation over land (Chen and Zhang, 2009), it appears that a more proper description of the canopy resistance over the desert steppe can be beneficial.

A major advantage of the HIRES simulation is that the precipitation distribution is much better represented compared to the LOWRES simulation. This is especially true for Europe where the simulations were verified against E-OBS data. Here, the HIRES simulation is much closer to the observed precipitation distribution although it also tends to produce spurious precipitation amounts. A reason for the overestimation of precipitation over Asia in the LOWRES simulation are the higher wind speeds at 10 m over eastern China and the Pacific (Fig. 13) leading to higher evaporation and thus a higher moisture availability (see also Fig. 10). As also the location of the subtropical high is changed, this can also lead to different precipitation patterns. The LOWRES simulation shows a similar overestimation of precipitation over India, Bangladesh, and Myanmar as the ECMWF operational model, probably related to the convection parameterization.

Over Central Asia, the benefit of the CP resolution is not clearly visible. This can be related to possible weaknesses in the CMORPH analysis over very complex terrain even when corrected with in-situ observations (Skok et al., 2016, e.g.). Another factor influencing the Indian Monsoon can be the role of aerosols but this is beyond the scope of this study. Over North America, the HIRES simulation shows a slightly worse correlation compared to the LOWRES experiment due to an overestimation of precipitation over the eastern United States (Fig. 14b) which apparently can be connected to a moist inflow bias at 925 hPa (Fig. 10b).

The overestimation of precipitation over the eastern part of the Unites States in both WRF simulations is related to the shift of the pressure system. This shift allows a moist inflow from the Atlantic and the Gulf of Mexico leading to higher precipitation amounts which are not simulated by the operational ECMWF forecasts.

## 6 Summary

Two latitude belt simulations with WRF between 20° N and 65° N, one at 0.12° resolution, one at the convection permitting resolution of 0.03° were analyzed for July and August 2013. Such high resolution simulations require computing resources of 84000 cores for 3.5 days and therefore are often limited to shorter periods or even case studies. Nevertheless, they are undisturbed by lateral boundary conditions at the western and eastern domain boundaries as in limited area model applications and therefore allow for new insights into model resolution dependence of the results. Further, since the results now depend on model physics and resolution only, the results can be assessed with respect to the model performance itself rather than the domain size and inconsistencies of model physics at the meridional boundaries. This is important since Eurasia and North America are characterized by the impact of the polar and subtropical jets and the sea surface temperatures of the Atlantic and Pacific Gyres, namely the Gulfstream and Kuroshio current on the general atmospheric circulation.

The simulations were compared to ECMWF operational analyses data at 0.12° resolution and to observational precipitation data sets of CMORPH and E-OBS at 0.25° resolution, since precipitation data is not available worldwide at higher resolution.

The objective of the study was to answer the questions posed in the introduction. The results are as follows:



1. What is the benefit of a CP resolution with respect to the spatial representation of large scale features in comparison to coarse resolution?

   An added value of the higher model resolution can be seen in the pressure fields of both months concerning the magnitude. The spatial distribution is not impacted by the resolution. The 925 hPa temperature shows an added value in the Pacific and Atlantic Ocean, but no added value is seen in the 925 hPa humidity.

2. Does the higher resolution lead to an improvement of surface variables such as 10-m wind speed and 2-m temperatures?

   Over land both simulations show the same biases in the 2m-temperature and 10m wind speeds. This indicates that the biases are subject to the physical parameterization schemes of WRF, namely those describing the sub-grid scale land-atmosphere feedback processes. Concerning the 10m wind speed HIRES shows an added value over the oceans.

3. What is the benefit of the CP resolution regarding the spatial distribution and amount of precipitation?

   HIRES shows an added value concerning the precipitation amount in the whole domain except Northern America, where LOWRES compares better with the studied precipitation data sets. In Europe HIRES results in an improved pdf of precipitation amounts. Concerning the spatial correlation no added value was gained from the HIRES simulation, however this is expected when comparing to the coarse observational data sets.

All in all the study reveals that a high resolution improves the general circulation, but is not sufficient to tackle the biases in long term simulations concerning the surface variables. Though computing resources are still growing, LAM will still be required for climate simulations. This study showed that the physical parameterizations need to be assessed to provide more accurate simulations of the climate and also to provide less biased surface variables to the impact models as required by the society. Namely the land-atmosphere feedback and interactions need to be investigated in a synergy of novel high resolution observational data (e.g. from the Surface-Atmosphere-Boundary-Layer-Exchange, Wulfmeyer and Coauthors, 2015) seamless model applications down to LES and new evaluation techniques (Wulfmeyer et al., 2016, e.g.) to improve the physical parameterization schemes on the applied model resolution.

**Code availability**

To download the WRF source code, users need to register on the following website: http://www2.mmm.ucar.edu/wrf/users/download/wrf-regist.php. Apart from the default required NetCDF and MPI libraries, users need to install the PNetCDF libraries version 1.5.0 or higher from the Argonne National Laboratory (https://trac.mcs.anl.gov/projects/parallel-netcdf).

**Appendix A: namelist.input used for the WRF simulations**

The following namelist.input was used for both simulations. For the LOWRES simulation, only the time step and grid resolution need to be changed.





```
     &time_control
       run_days                    = 0,
       run_hours                   = 0,
       run_minutes                 = 0,
 run_seconds                 = 0,
        start_year                 = 2013
       start_month                 = 07
       start_day                   = 01
       start_hour                  = 0
start_minute                = 0
       start_second                = 0
       end_year                    = 2013
       end_month                   = 09
       end_day                     = 01
end_hour                    = 0
       end_minute                  = 0
       end_second                  = 0
        interval_seconds           = 21600,
        input_from_file            = . true .
history_interval           = 30,
       frames_per_outfile          = 1
        restart                    = . false .,
        restart_interval           = 720,
        override_restart_timers    = . true .
io_form_history             = 11,
        io_form_restart            = 11,
       io_form_input               = 102,
       io_form_boundary            = 11,
       io_form_auxinput1           = 11,
debug_level                 = 0,
       nocolons= . true .
       io_form_auxinput4           = 11
       auxinput4_inname            = "wrflowinp_d<domain>"
        auxinput4_interval         = 360
auxhist23_outname='wrfpress_d<domain>_<date>'
```





```
        io_form_auxhist23  = 11
         auxhist23_interval   = 30,
        frames_per_auxhist23  = 1
         diag_print =1,
auxhist2_outname='afwa_d<domain>_<date>'
        io_form_auxhist2  = 11
         auxhist2_interval   = 15,
        frames_per_auxhist2  = 1
         use_netcdf_classic =. true .
10      /

     &diags
     p_lev_diags                         = 1
      num_press_levels                   = 7
press_levels                       = 92500, 85000, 70000, 50000, 30000, 20000, 10000
      use_tot_or_hyd_p                   = 2
     /
     &domains
     time_step                           = 10
time_step_fract_num                 = 0
      time_step_fract_den                = 1
     max_dom                             = 1
     s_we                                = 1
     e_we                                = 12000
s_sn                                = 1
     e_sn                                = 1500
      s_vert                             = 1
     e_vert                              = 57
     eta_levels  =    1.000,0.997,0.993,0.989,0.983,0.972,0.962,0.952
30      ,0.942,0.932,0.917,0.903,0.889,0.875,0.852,0.826,0.799,0.771,
        0.748,0.725,0.7,0.678,0.653,0.628,0.590,0.557,0.515,0.480,
        0.445,0.410,0.375,0.340,0.305,0.280,0.25,0.219,0.191,0.174,
        0.157,0.142,0.128,0.114,0.102,0.091,0.080,0.070,0.061,0.052
        ,0.044,0.037,0.030,0.024,0.018,0.013,0.008,0.003,0.000,
num_metgrid_levels                  = 138,
```



|  |  |  |
|---|---|---|
| p_top_requested | = 1000, | |
| dx | = 3335.324, | |
| dy | = 3335.324, | |
| grid_id | = 1, | |
| 5 | parent_id | = 1, |
|  | i_parent_start | = 1, |
|  | j_parent_start | = 1, |
|  | parent_grid_ratio | = 1, |
|  | parent_time_step_ratio | = 1, |
| 10 | feedback | = 1, |
|  | smooth_option | = 0, |
|  | use_surface | = . false ., |
|  | sfcp_to_sfcp | = . false . |
|  | use_adaptive_time_step | = . false . |
| 15 | step_to_output_time | = . true . |
|  | target_cfl | = 1.3, |
|  | max_step_increase_pct | = 50, |
|  | starting_time_step | = −1, |
|  | max_time_step | = 15, |
| 20 | min_time_step | = 1, |
|  | / | |
|  | | |
|  | &physics | |
|  | sst_update | = 1, |
| 25 | mp_physics | = 10 |
|  | ra_lw_physics | = 4 |
|  | ra_sw_physics | = 4 |
|  | radt | = 3 |
|  | sf_sfclay_physics | = 1 |
| 30 | sf_surface_physics | = 2, |
|  | bl_pbl_physics | = 1, |
|  | bldt | = 0, |
|  | topo_wind | = 1 |
|  | cu_physics | = 0, |
| 35 | cudt | = 0, |



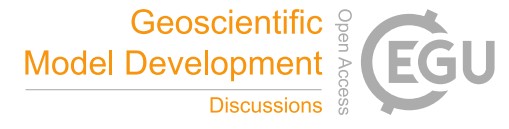

```
      kfeta_trigger                 = 2,
      isfflx                        = 1,
      ifsnow                        = 1,
      icloud                        = 1,
surface_input_source          = 1,
      num_soil_layers               = 4,
      mp_zero_out                   = 0,
      sf_urban_physics              = 0,
      maxiens                       = 1,
maxens                        = 3,
      maxens2                       = 3,
      maxens3                       = 16,
      ensdim                        = 144,
      slope_rad                     = 0,
topo_shading                  = 0,
      num_land_cat                  = 21,
      iz0tlnd                       = 1,
      shcu_physics                  = 3
      sf_ocean_physics              = 0
usemonalb = . true .
      do_radar_ref  = 1,
      hail_opt      = 1,
      /

&afwa
      afwa_diag_opt=1
      afwa_severe_opt=1
      afwa_ptype_opt=1
      afwa_radar_opt=1
afwa_vis_opt=1
      afwa_cloud_opt=1
      /

      &dynamics
w_damping                     = 1,
```





```
    diff_opt                        = 1,
    km_opt                          = 4,
    gwd_opt                         = 0,
    diff_6th_opt                    = 2,
diff_6th_factor                 = 0.12,
    base_temp                       = 290.
    damp_opt                        = 3,
    zdamp                           = 5000.,
    dampcoef                        = 0.2,
khdif                           = 0,
    kvdif                           = 0,
    non_hydrostatic                 = . true .,
    moist_adv_opt                   = 1,
    scalar_adv_opt                  = 1,
epssm           = 0.5
    /

    &bdy_control
spec_bdy_width                  = 5,
    spec_zone                       = 1,
    relax_zone                      = 4,
    specified                       = . true .,
    nested                          = . false .,
periodic_x                      = . true .
    /

    & namelist_quilt
    nio_tasks_per_group  = 0,
nio_groups = 1,
    /
```

*Acknowledgements.* We are grateful to the High-Performance Computing Center Stuttgart (HLRS) and to Cray Inc. for providing the tremendous amount of computing time required for this simulation on XC40 system. Special thanks goes to U. Küster, T. Beisel, and T. Bönisch



from HLRS and to S. Andersson and S. Dieterich from Cray Inc. . We are also grateful to the ECMWF for providing operational analysis data.



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





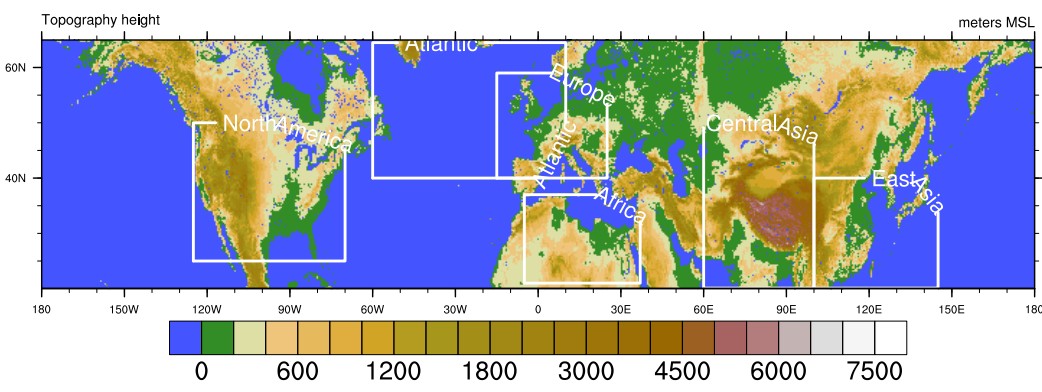

**Figure 1.** Model domain of the latitude belt simulation. The white rectangles denote the domains used for verification of precipitation.



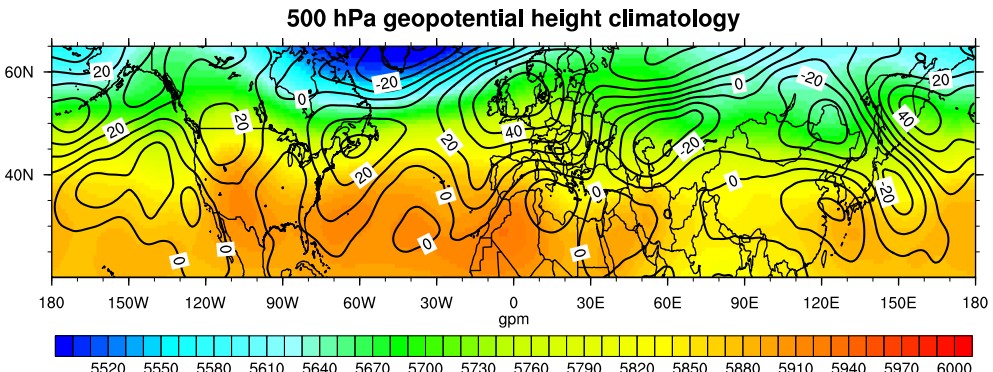

**Figure 2.** ERA-INTERIM 500 hPa geopotential height climatology for July and August 2013 (shaded). The contour lines show the anomaly during July and August 2013 compared to the period 1979–2012.





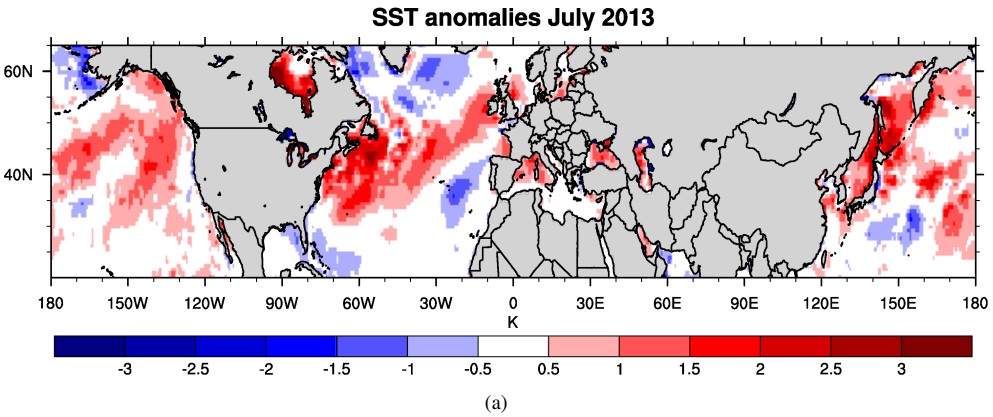

(a)

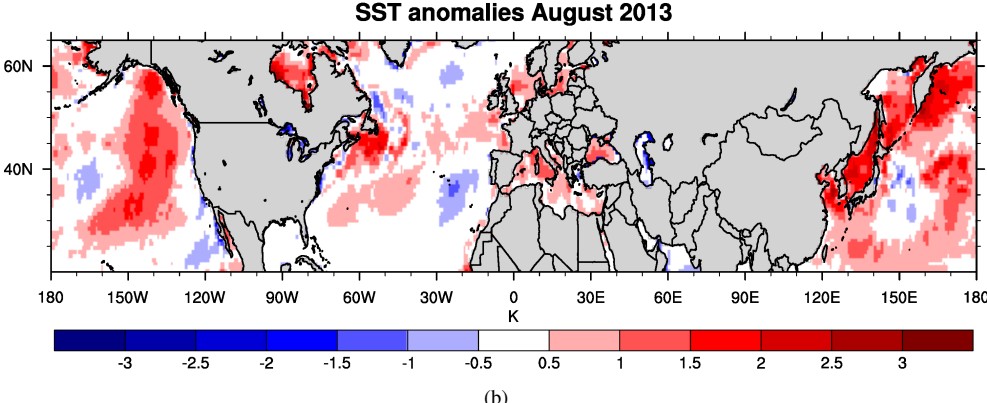

(b)

**Figure 3.** ERA-INTERIM sea surface temperature (SST) anomalies in July 2013 (upper panel) and August 2013 (lower panel) compared to the climatological period 1979–2012.





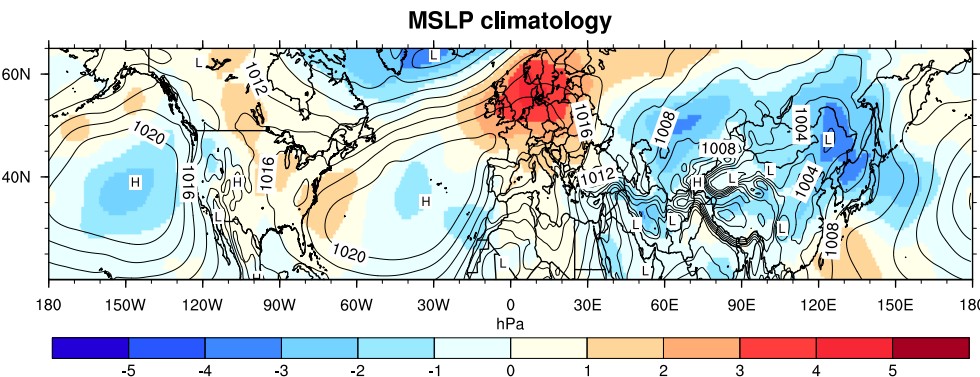

**Figure 4.** Mean sea level pressure climatology for July and August 2013 (solid lines) from ERA-INTERIM. The shaded areas show the anomaly during July and August 2013 compared to the period 1979–2012.







**Figure 5.** Average mean sea level pressure at 12 UTC for July 2013 (left column) and August 2013 (right column). The first row shows the ECMWF analysis followed by the LOWRES simulation (c,d) and the HIRES simulation (e,f). The two lowermost rows show the mean bias for the LOWRES (g,h) and the HIRES simulations (i,j) compared to the ECMWF analysis for the two different months.





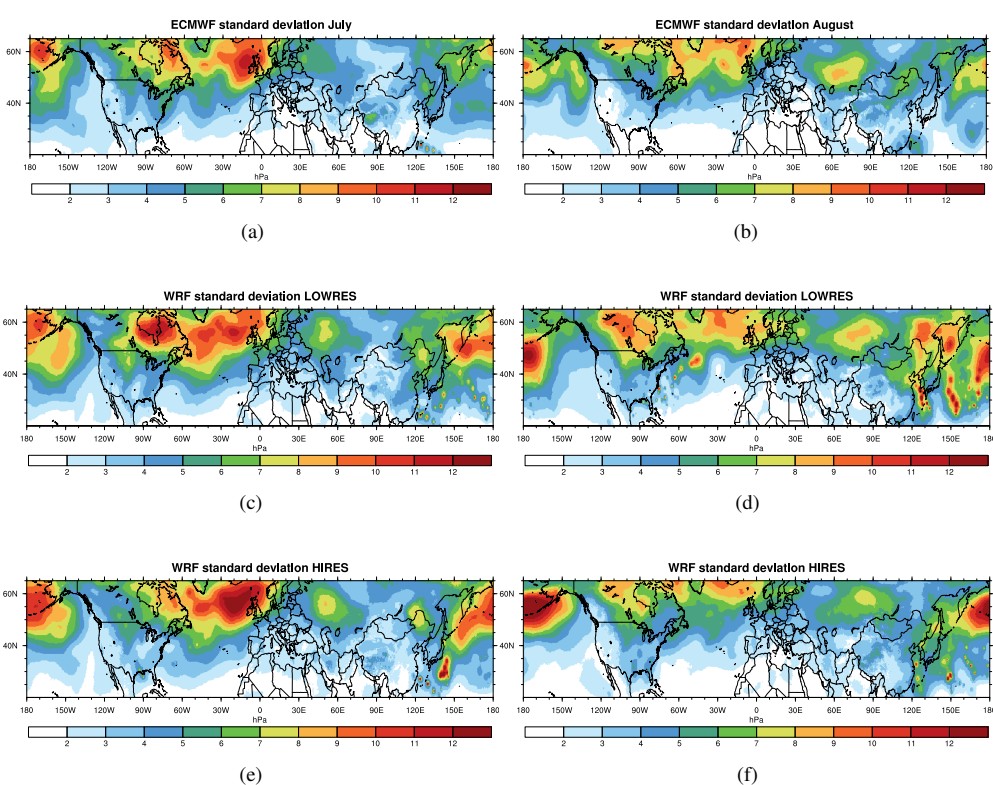

**Figure 6.** Mean sea level pressure standard deviation at 12 UTC for July 2013 (left column) and August 2013 (right column). The top row shows the ECMWF analysis, the middle row shows the LOWRES simulation and the bottom row displays the HIRES simulation.





**Figure 7.** Mean 300 hpa wind velocities for July 2013 (left column) and August 2013 (right column).The top row shows the ECMWF analysis, the middle row shows the LOWRES simulation and the bottom row displays the HIRES simulation.





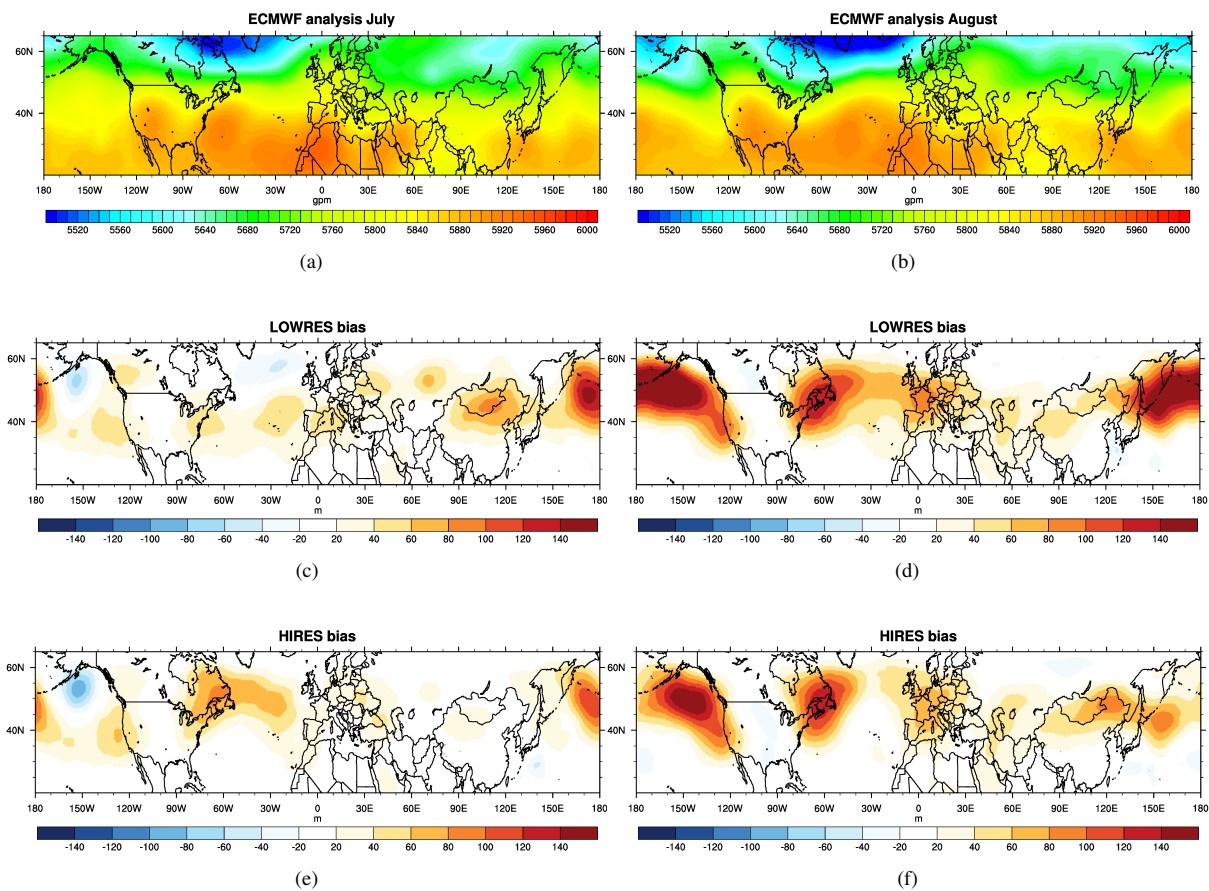

**Figure 8.** Mean 500 hPa geopotential height and mean differences between the WRF simulations and ECMWF analysis for July 2013 (left column) and August 2013 (right column).The top row show the ECMWF analysis, the middle row show the LOWRES simulation and the bottom row display the HIRES simulations.





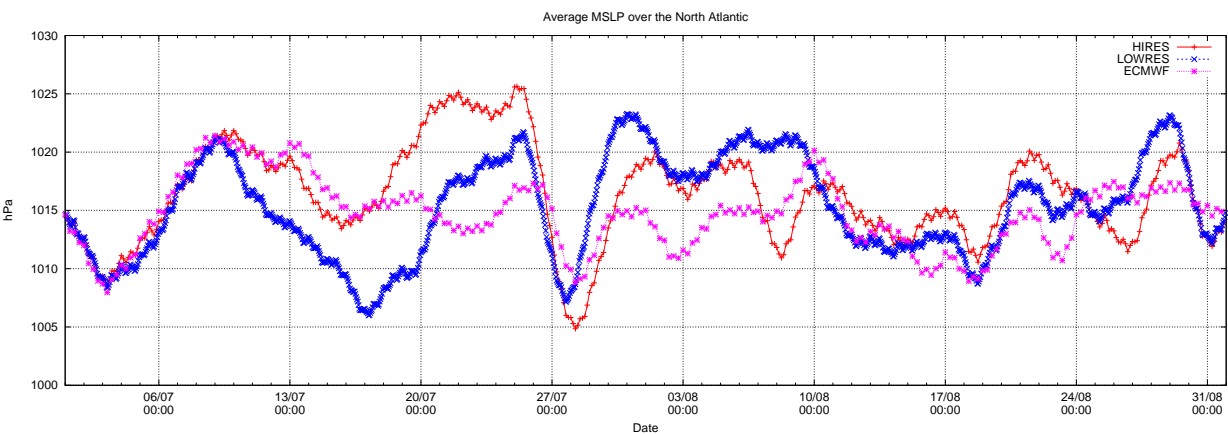

**Figure 9.** Time series of the MSLP averaged between 40° N and 65° N, and 60° W and 10° E.





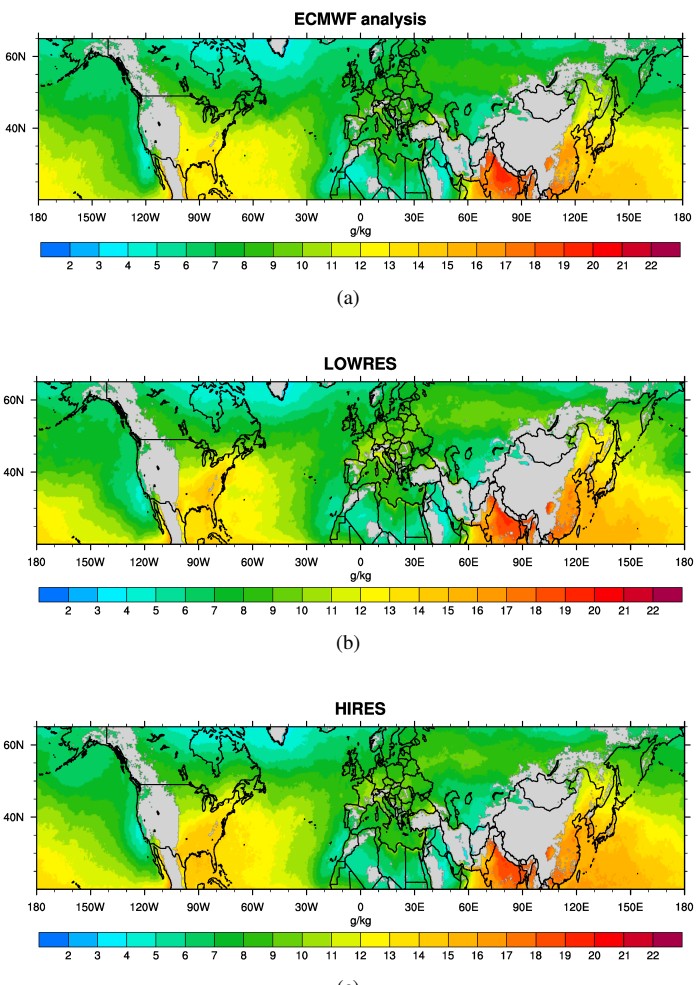

**Figure 10.** Average 925 hPa water vapor mixing ratio for 12 UTC of the ECMWF analysis (a) and the LOWRES simulation (b). The bottom panel shows the HIRES simulation.



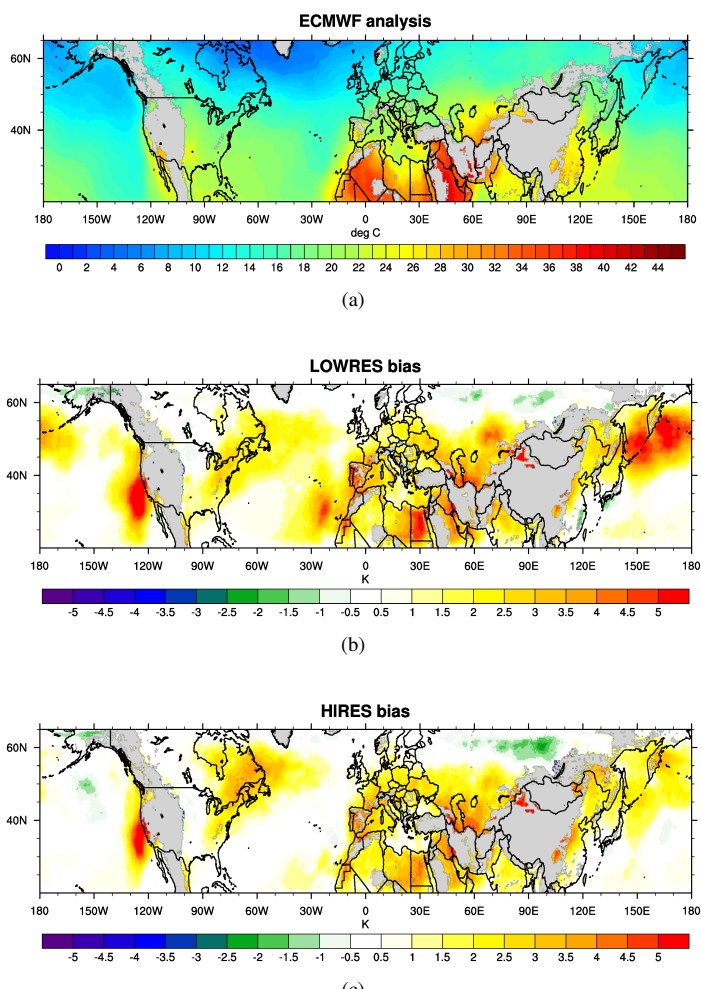

**Figure 11.** Mean 925 hPa temperature for 12 UTC of the ECMWF analysis (a). (b) and (c) show the deviation of the LOWRES and HIRES simulation from the ECMWF analysis, respectively.





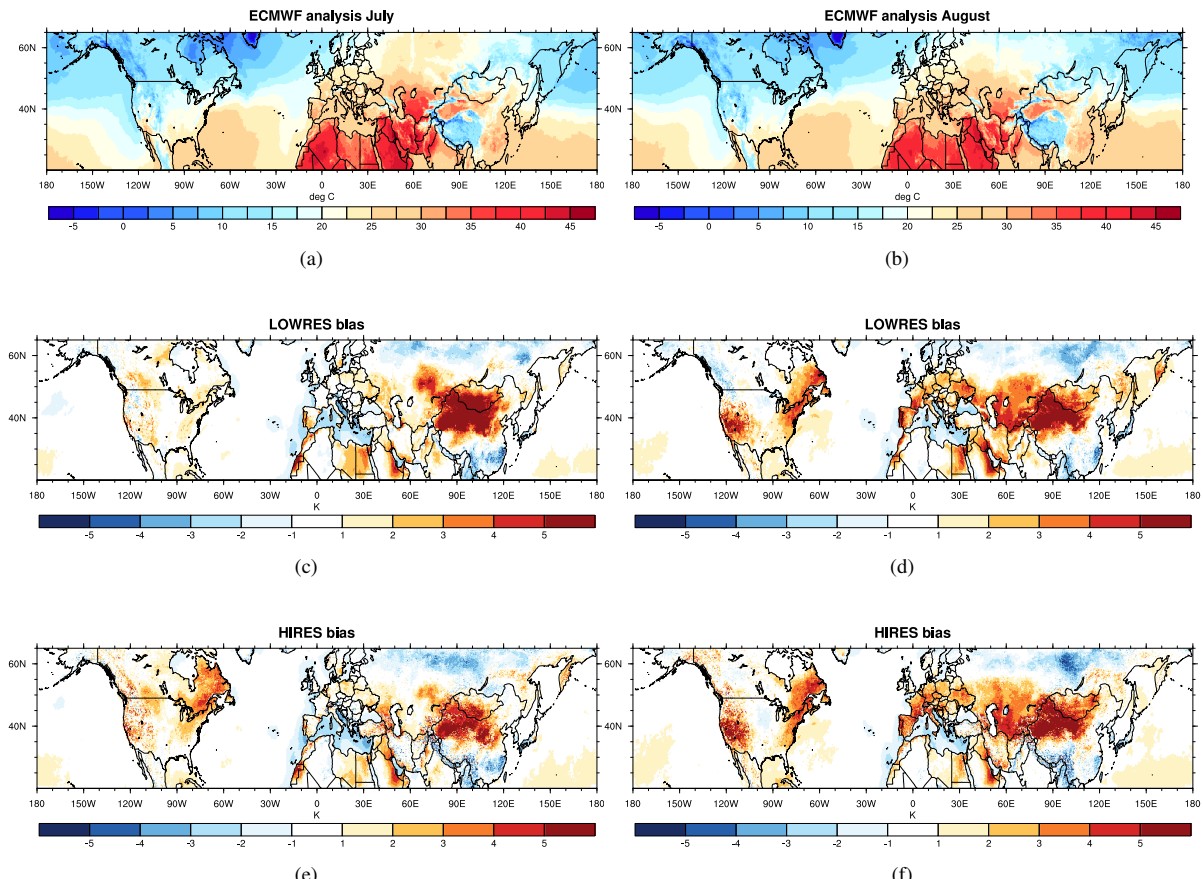

**Figure 12.** Mean 2-m temperature at 12 UTC in July (left column) and August (right column). The top row shows the ECMWF analysis, the middle row displays the LOWRES simulation and the bottom row represents the HIRES simulation. Reddish colors indicate a warm bias of the WRF simulations.





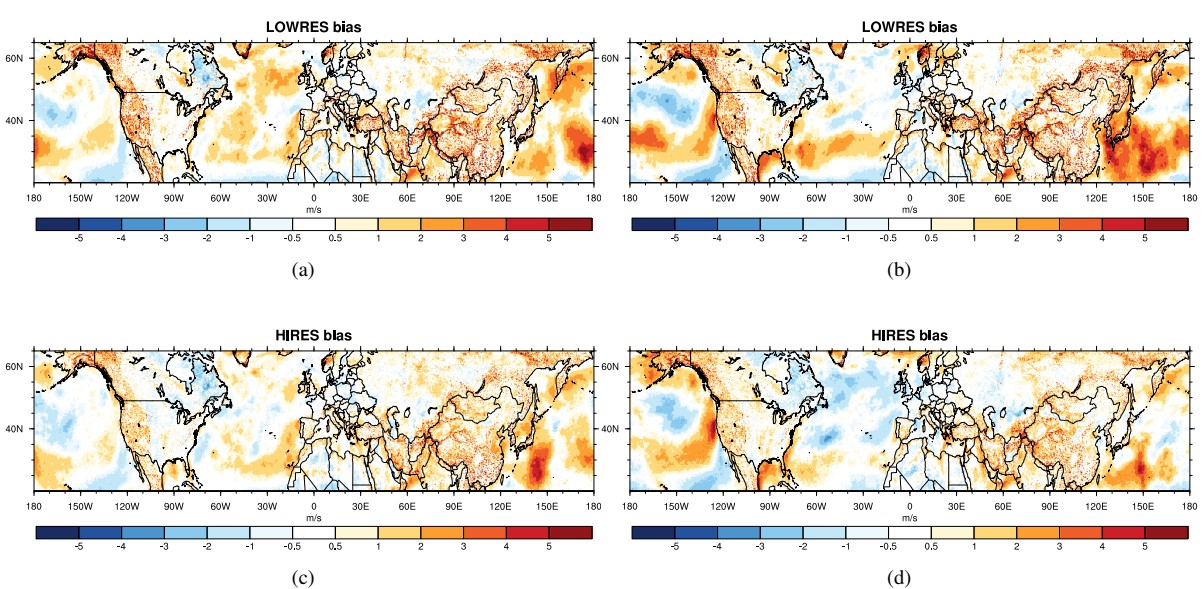

**Figure 13.** 10-m wind speed bias in July (left column) and August 2013 (right column) for the LOWRES simulation (upper row) and the HIRES simulation (lower row).





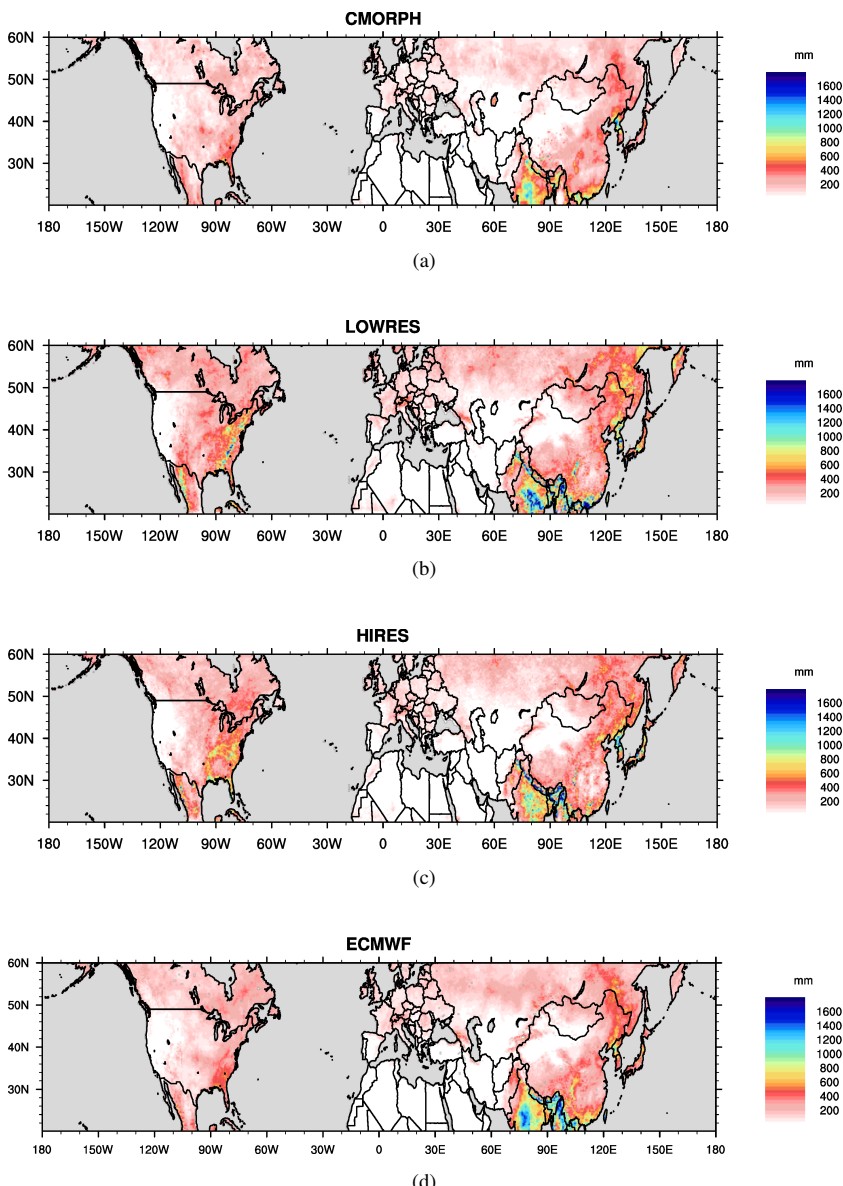

**Figure 14.** 2 month accumulated precipitation. (a) shows the CMORPH analysis, (b) shows the LOWRES simulation and (c) displays the HIRES simulation. (d) displays the accumulated precipitation from the operational ECMWF 12 h forecast started at 00 UTC and 12 UTC each day.





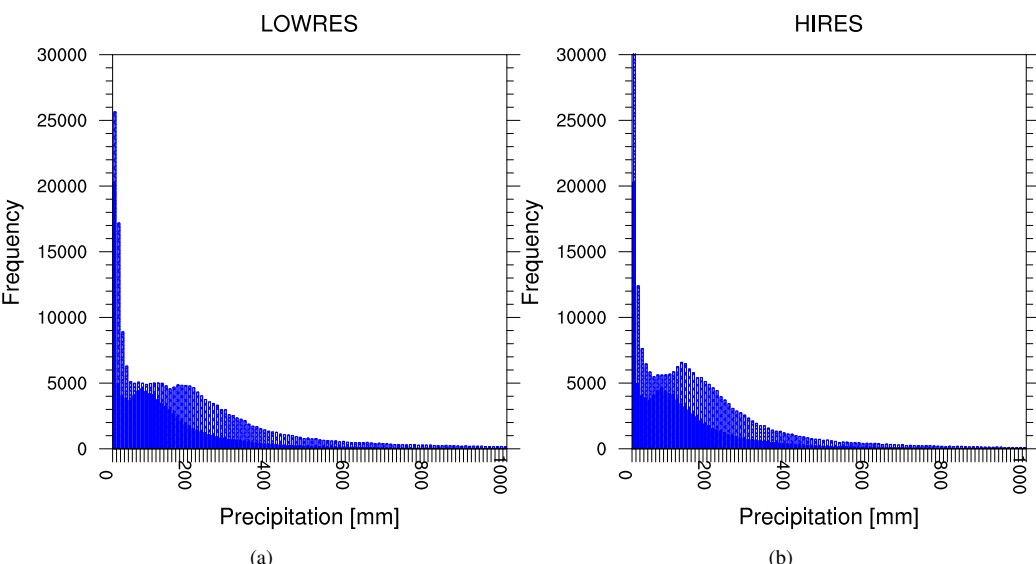

**Figure 15.** Histogram of the 2 month accumulated precipitation by using the CMORPH data over land points only. The filled blue bars denote the CMORPH data set and the cross-hatched bars denote the LOWRES (left) and HIRES simulation data (right).



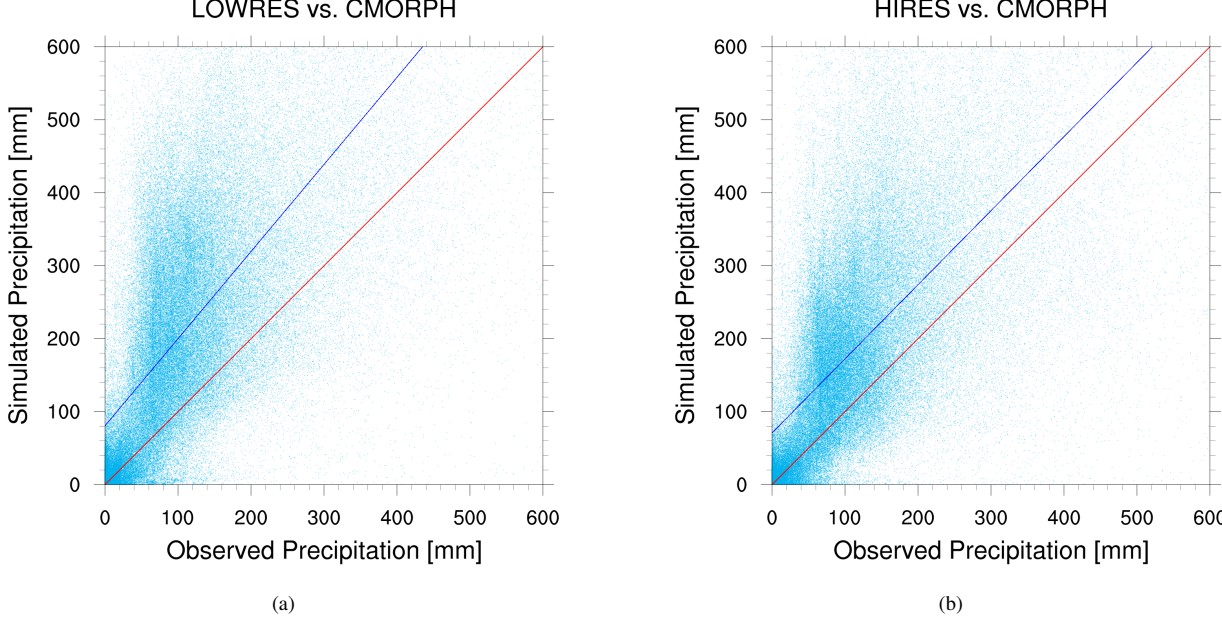

(a)           (b)

**Figure 16.** Scatter plot of the accumulated precipitation over the two month period including regression lines. The left panel shows the LOWRES simulation vs. the CMORPH data. The red line would be the perfect result. The right panel shows the HIRES simulation vs. the CMORPH data.





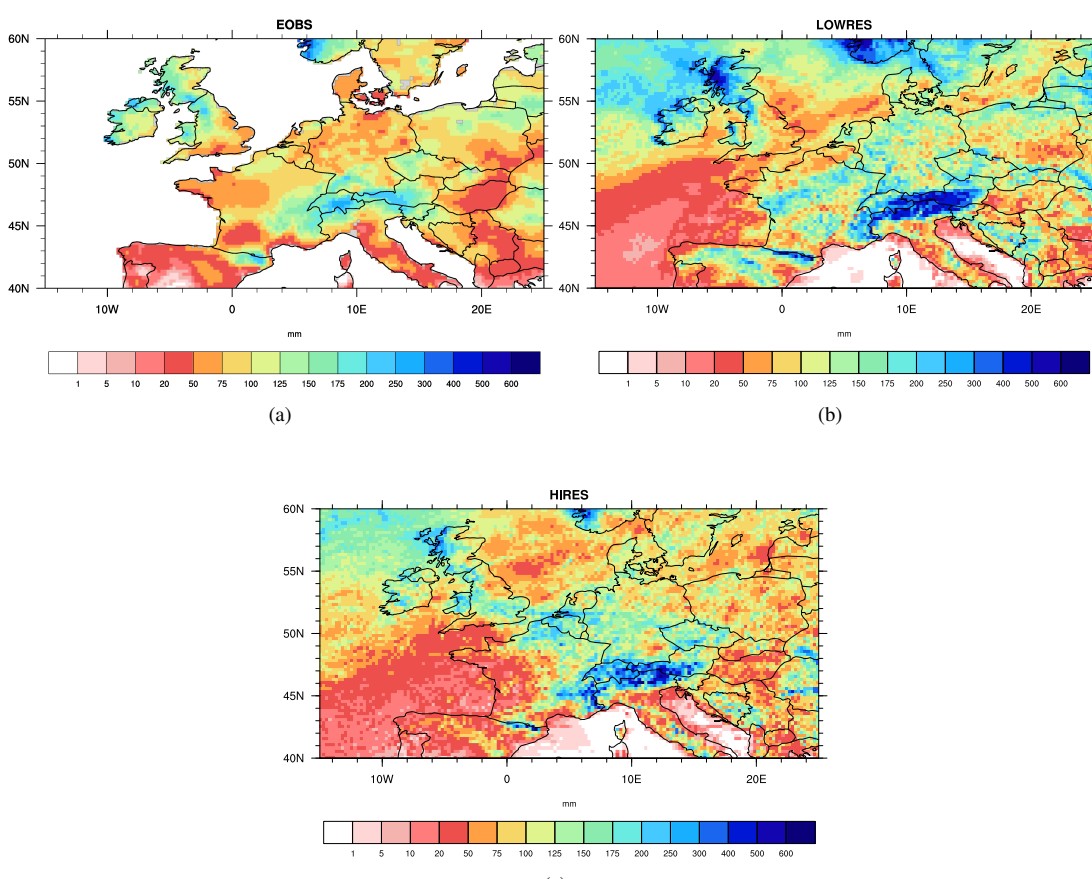

**Figure 17.** 2 month accumulated precipitation over Europe. (a) shows the E-OBS data set, (b) shows the LOWRES simulation, and (c) displays the HIRES simulation.



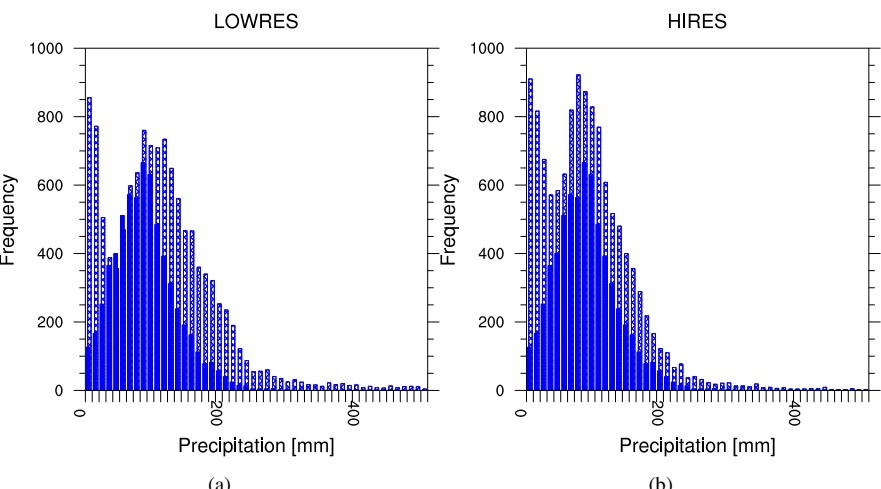

**Figure 18.** Histogram of the 2 month accumulated precipitation over Europe using the E-OBS data set. The filled blue bars denote the observation data set and the cross-hatched bars denote the HIRES (left column) and LOWRES simulation data (right column).

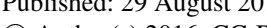

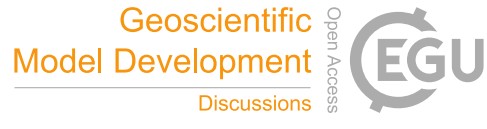

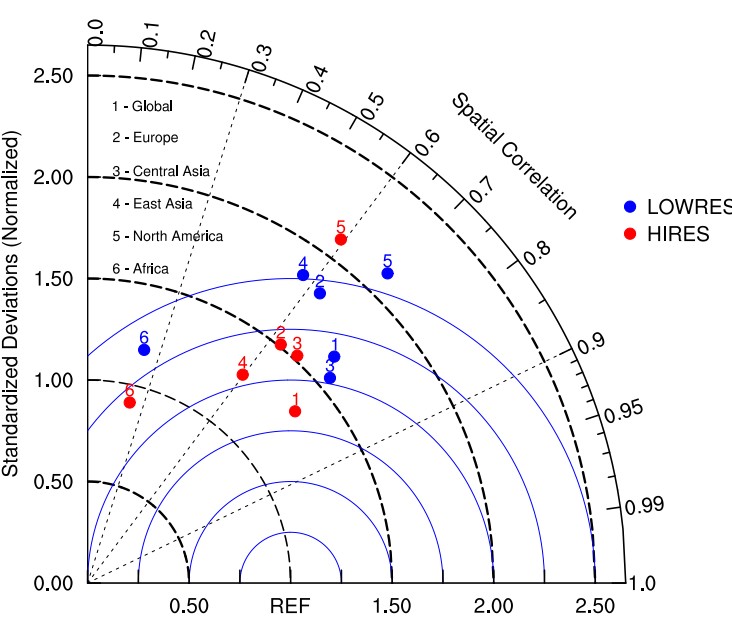

**Figure 19.** Taylor diagram of the accumulated precipitation over land points. The simulations were plotted against CMORPH observations, except for Europe where the E-OBS data set is the reference.





**Table 1.** Physics parameterizations used in the WRF simulations.

| Parameterization | Scheme | Reference |
|---|---|---|
| Cloud microphysics | Morrison 2-moment | Morrison et al. (2009) |
| Radiation | RRTMG | Iacono et al. (2008) |
| PBL | YSU | Hong (2007) |
| Shallow convection | GRIMS | Hong et al. (2013) |
| Cumulus parameterization | KF-ETA | Kain (2004) |
| Surface layer | MM5 scheme | Jimenéz et al. (2012) |
| Land Surface | NOAH LSM | Ek et al. (2003) |