# Peer review of "Continuous high resolution mid-latitude belt simulations for July-August 2013 with WRF"

_Geoscientific Model Development, 2016_

## Referee Comment (RC1) · Anonymous Referee #1 · 9 Nov 2016

The authors have driven the atmospheric model WRF with analysis fields of the operational forecasting system of the ECMWF to simulate the period July to August 2013 in a mid-latitudinal belt (20°N to 65°N) around the globe with two grid sizes: 0.12° and 0.03°. Both simulations make use of sea surface temperatures from the Operational Sea Surface Temperature and Sea Ice Analysis (OSTIA) data. In the 0.03° simulation, parameterisation for deep convection is turned off. In order to evaluate the belt simulations, they are compared to various reference datasets (analysis fields of the operational forecasting system of the ECMWF, EOB-S, CMORPH) and in several subdomains. In addition, the authors ask specific questions about the added value in the 0.03° simulation.

[Figure]

General Comments

The paper combines two relatively new innovations, convection permitting simulations and belt simulations. The strengths and weaknesses of both innovations are largely unknown and hence, the paper is worth to be published. However there are two major issues that need to be clarified first:

(1) Scientific quality

The solution of a local area model is partly predominated by its lateral boundary conditions (LBCs). The larger the model domain becomes, the weaker becomes the coupling to its LBCs and the larger become large-scale deviations from its driving data in the interior of the model. Kida et al. (1991) and Paegle et al. (1996) are often cited in this context. More recently, Becker et al. (2015) demonstrated that a local area model creates artificial flows to compensate those large-scale deviations in order to achieve physical consistency with the LBCs along the lateral boundaries and that an increase of the model domain does not change this – the artificial flows simply become more complex.

In the presented belt simulations, there are no western and eastern boundaries and hence, the decoupling becomes an important factor. This can be seen in principal in Fig. 9: the model creates significant anomalies in MSLP (low and high pressure systems are created that do not exist and vice versa) in the Atlantic region, but from time to time (e.g. around July 27 and August 10 to 15) the influence of the driving data becomes dominant. The fact that there is some coupling to the LBCs at all comes from the location of the sub-domain: the Atlantic region touches the northern boundary. In the interior of the model domain, the coupling might be much smaller.

Hence, the simulations may be affected by large-scale decoupling to such a large extent that the entire evaluation in its current stage is flawed. Shifts in time/space between modelled and observed phenomena are limiting the applicability of traditional statistical analysis. Biases and other error measures are showing the summary ef-

fect of large-scale decoupling and model deficiencies (which should be the only focus in a model evaluation study). A common approach to overcome this mismatch is to extend the simulation period to multiple decades and evaluate statistical measures in a climatological way (as it is done for climate models, for instance). However, in the face of high computational costs, this might not be feasible. For the purpose of model evaluation it would be enough to demonstrate that the simulations are lying within the bandwidth of possible realistic developments. The climatological year-to-year variability (on a monthly basis), which could be derived from ERA-Interim, or extended ensemble forecast data (e.g. http://www.ecmwf.int/en/forecasts/datasets/set-vi) – as the forecast model runs without ingestion of observation – could be used to define such space of possible developments.

Because of the large-scale decoupling that makes the simulations partly incomparable to observational or observation based data, the investigation of added value is flawed, too. In addition, the asked questions about added value are way too generally expressed. With one coarse and one fine resolved simulation of the same model, no robust conclusion on added value can be drawn. In such a case, the added value analysis is limited to this specific case. By the way, for demonstrating added value, it is not enough to show that biases on a monthly basis are reduced, because monthly biases are the result of multiple processes and phenomena that may take place at the same time and also in sequences. So, a reduction of a monthly bias can be the result of enlarged process and phenomena related biases that are simply cancelling out each other. Hence, demonstrating added value includes a thorough investigation of the underlying processes and phenomena plus a demonstration that these processes and phenomena are more properly captured by the finer resolved model. To solve this issue, the authors could either include such a thorough process and phenomena based analysis or should put more effort on the model evaluation and its problematic (see above) and do not announce added value in such a prominent way.

Since large-scale decoupling plays such an important role, it needs to be an integrative

Interactive
comment

part of the discussion (Section 5) and summary (Section 6) sections.

Becker N., U. Ulbrich, R. Klein (2015), Systematic large-scale secondary circula-
tions in a regional climate model. Geophys. Res. Lett., 42, 4142–4149. doi:
10.1002/2015GL063955. Kida, H., T. Koide, H. Sasaki, and M. Chiba (1991), A new
approach for coupling a limited area model to a GCM for regional climate simulations.
J. Meteor. Soc. Japan, 69, 723–728. Paegle, J., K. C. Mo, and J. N. Paegle (1996),
Dependence of simulated precipitation on surface evaporation during the 1993 United
States summer floods. Mon. Wea. Rev., 124, 345–361.

(2) Presentation quality

Reference data, error measures, including an explanation why theses reference
datasets and error measures are selected and how data from different grids is
remapped onto a common evaluation grid is missing in the experimental setup (Sec-
tion 2). Instead this information is (partly) given at other places, for instance at the
beginning of the result section (Section 4). Having an evaluation concept in section 2
summarising all of this would increase the readability of the manuscript.

Specific Comments

The following study would nicely fit into the introduction section:

Prein, A. F., A. Gobiet, H. Truhetz, K. Keuler, K. Goergen, C. Teichmann, C. Fox Maule,
E. van Meijgaard, M. Déqué, G. Nikulin, R. Vautard, A. Colette, E. Kjellström, and
D. Jacob (2015), Precipitation in the EURO-CORDEX $0.11°$ and $0.44°$ simulations:
High resolution, high benefits?, Climate Dynamics, 46, 383–412, 10.1007/s00382-015-
2589-y

Page 3, lines 14, 15: It is unclear how errors in the large scale circulation patterns can
be traced back to the applied physics schemes, especially in the light of large-scale
decoupling.

Page 3, lines 22, 23: It is not clear how large uncertainties over the Atlantic and Pacific

can be explained by differences in model physics, especially in the light of large-scale decoupling which is also active when model physics are identical (simply extending the model domain of a local area model by some grid cells into one direction gives different results; see Becker et al., 2015).

Page 5, line 14: What is the advantage of using OSTIA instead of SST from the operational ECMWF analyses? OSTIA is given on a daily resolution and need to be interpolated in time (see page 5 line 34 to page 6, line 3), while SST from ECMWF is already on a 6 h basis. I am not an SST expert, but a short literature research brought up a paper from Seo et al. (2014) which demonstrates the importance of sub-daily SST variability to properly capture the onset and intensity of Madden–Julian oscillation (MJO) convection in the Indian Ocean in a coupled WRF-ocean model. Seo, H., A. C. Subramanian, A. J. Miller, and N. R. Cavanaugh (2014), Coupled Impacts of the Diurnal Cycle of Sea Surface Temperature on the Madden–Julian Oscillation. J. Climate, 27, 8422–8443, doi: 10.1175/JCLI-D-14-00141.1.

Page 6, lines 11 to 17: the 0.03° and 0.12° simulations make use of pnetcdf. A discussion about pnetcdf is missing here. Does pnetcdf solve the problem?

Page 8, line 25: What is "good agreement"? What biases are acceptable? (These questions should be tackled in an evaluation concept in section 2.)

Page 11, lines 7 to 8: There must be something fundamentally going wrong with the model in this specific region. Maybe it is related to the initialisation of the soil. The authors are encouraged to contact WRF experts (e.g. Walter Immerzeel, University of Utrecht, or the CORDEX-South-Asia or CORDEX-Central-Asia communities) that are operating the model in this region. There is also a new reference dataset for temperature available. It is called WFDEI (Weedon et al., 2010; 2011) and can be downloaded from ftp://rfdata:forceDATA@ftp.iiasa.ac.at

Weedon, G.P., Gomes, S., Viterbo, P., Österle, H., Adam, J.C., Bellouin, N., Boucher, O., and Best, M., 2010. The WATCH Forcing Data 1958-2001: a meteorological forcing dataset for land surface- and hydrological models. WATCH Tech. Rep. 22, 41p (available at www.eu-watch.org/publications ).

Weedon, G.P., Gomes, S., Viterbo, P., Shuttleworth, W.J., Blyth, E., Österle, H., Adam, J.C., Bellouin, N, Boucher, O., and Best, M., 2011. Creation of the WATCH Forcing data and its use to assess global and regional reference crop evaporation over land during the twentieth century. J. Hydrometerol. 12, 823-848, doi: 10.1175/2011JHM1369.1.

This dataset is based on ERA-Interim, but it corrects temperatures in a way that it is consistent with the observed behaviours of glaciers in the Himalayan region. It might be a more reliable reference dataset than the ECMWF analysis fields.

Technical Corrections

Page 3, line 16: Is there a reference for the storm systems affecting Europe?

Page 3, line 33: typo – "This study is organised . . ."

Page 4, line 14: Euro-CORDEX should be referenced by Jacob et al. (2014).

Jacob, D., J. Petersen, B. Eggert, A. Alias, O. B. Christensen, L. M. Bouwer, A. Braun, A. Colette, M. Déqué, G. Georgievski, E. Georgopoulou, A. Gobiet, L. Menut, G. Nikulin, A. Haensler, N. Hempelmann, C. Jones, K. Keuler, S. Kovats, N. Kröner, S. Kotlarski, A. Kriegsmann, E. Martin, E. van Meijgaard, C. Moseley, S. Pfeifer, S. Preuschmann, C. Radermacher, K. Radtke, D. Rechid, M. Rounsevell, P. Samuelsson, S. Somot, J.-F. Soussana, C. Teichmann, R. Valentini, R. Vautard, B. Weber, and P. Yiou (2014), EURO-CORDEX: New high-resolution climate change projections for European impact research, Regional Environmental Change, 14, 563–578, 10.1007/s10113-013-0499-2.

Page 5, line 18: Are there any references for the studies that have shown the spin-up time for NOAH's land surface model?

Page 9, line 1 to 2: Is there a reference to the analysis on tropical storms of JMA?

Page 9, line 5: Is there a reference to this jet stream north of the Tibetan Plateau which is typical for the summer monsoon?

Page 9, line 10: up to now, it was not clear that RMSE is used at all. It is also not clear which RMSE is used (the RMSE of monthly means or on a daily basis or whatsoever)

Page 10, line 8: typo – ". . . due to small . . ."

Page 10, line 19: typo – ". . . caused by the inaccurate . . ."

Page 15, line 29 to 30: cumulus parameterisation is also changed in LOWRES. Is GRIMS also active in LOWRES?

Figure 2: colours of the shades are too intensive, continents can only hardly be seen. Also the structure of the plot should be consistent with Figure 4 (anomalies should be shaded and the climatology should be in solid lines).

———————————

---

## Referee Comment (RC2) · Anonymous Referee #2 · 25 Nov 2016

The authors investigate the benefits of convection permitting modeling by employing the WRF modeling system in a channel configuration over the Northern Hemisphere (between 20° N and 65° N) at resolutions of 0.12° (LOWRES) and 0.03° (HIRES), respectively. In HIRES the deep convection parameterization scheme is turned off. The necessarily short integration period covers the summer of 2013 (July and August). This period was notable for exhibiting a strongly positive phase of the North Atlantic Oscillation and generally weaker subtropical highs over the Atlantic and Pacific basins. The driving data is from the ECMWF operational analysis data at the northern and southern boundaries and the OSTIA 5km SST data set at the sea surface. The authors then compare their results to reanalysis and gridded combined data products (e.g., E-OBS,

CMORPH). They aim to answer the three questions: 1) What is the benefit of a CP resolution with respect to the spatial representation of large-scale features in comparison to coarse resolution? 2) Does the higher resolution lead to an improvement of surface variables such as 10m windspeed and 2m temperatures? 3) What is the benefit of the CP resolution with respect to the spatial distribution and amount of precipitation?

Given large channel domain and the fact that convective permitting simulations are still relatively rare this study can potentially make a useful contribution to our understanding of how and why simulations at these grid spacings are useful. However, the experiment is not designed in such a way that it can answer the questions as they are posed. The shortcomings are detailed below as are suggests for improvement. I will focus on what I see as the two major issues that must be addressed before the manuscript can move forward. There are likely more specific comments but these can be addressed in the next revision.

General comments

Due to the channel set up the model simulations are largely "free". In other words internal variability can account for much of the difference that we see between the simulations and the reference data sets. In fact there is little reason to assume that they would in anyway resemble each other. In the absence of nudging the one-to-one comparison of the model simulations with the reference fields for the large-scale circulation is doomed to fail. This is illustrated most clearly through examination of the Figures 4 and 5. The dominant anomalies in the large-scale circulation for 2013 are a weakening of the subtropical highs over the ocean basins and a strengthening of the low-pressure anomalies over the Eurasia. The so-called model biases wipe out this weakening over the subtropical highs and intensify the low-pressure anomalies over the Eurasia. From there the rest of the comparisons are uninformative at best. Therefore, I would suggest the authors focus on whether this type of simulation is fit for purpose. In other words, can the model perform the task for which it is intended and does the HIRES simulation perform this task better, or more accurately than the LOWRES simulation? One way

the authors could do this would be to show, via hatching for example, areas where the modeled field falls outside the +/- 2 standard deviation confidence bounds of the observations. Given that these simulations are basically single realizations of internal variability, weakly constrained by the lower and north/south boundaries this is a more fair and appropriate comparison. Another solution would be to re-run the experiments, but constrain the flow so that expectation could be that the model, in the absence of internal model errors, would reproduce the temporal evolution of the weather over the course of July and August 2013.

The other issue relates to added value. Given the issues described above and the face that this is a single model, case study experiment, it is very, very, difficult to convincingly argue for added value. Rather than focus on added value using such measure as Taylor diagrams and RMSE, the authors could perhaps focus more on processes that are more accurately captured in the HIRES simulation. Examples are diurnal cycles of winds and precipitation, blocking associated with heat waves, etc. The authors should not focus on spatial comparisons as there is little reason to expect high spatial correlation between the simulations and the reference data other than that due to the fact there are climatological patterns that the simulations will somewhat follow. If the authors are really set on showing added value then I would recommend they use something like the Perkins skill score which assesses the similarity of two pdfs (Perkins et al. 2007). This metric is quite a bit more informative than the approaches shown used in the manuscript, which rely heavily on visual inspection.

Perkins, S. E., Pitman, A. J., Holbrook, N. J., & McAneney, J. (2007). Evaluation of the AR4 climate models' simulated daily maximum temperature, minimum temperature, and precipitation over Australia using probability density functions. Journal of climate, 20(17), 4356-4376.

Presentation Quality

Some context for the study is lacking. Why is summer 2013 chosen? Some information

on datasets and calculations is missing. The resolution of OSTIA is about 5km. This does not appear in the text. Some of the reference data sets are described in the experiment set up section. Some description only comes later in the results. It is a bit confusing. Better would be to describe all the reference data sets their strengths and weaknesses, resolution, etc. in a subsection of the experimental set up.

Specific comments The abstract is much too long and without critical insight. The abstract should not just be a laundry list of the results but a brief exposition of key findings. The reader should immediately grasp why this paper is of interest. The contribution this study is making should come through in the abstract.

Page 5 L3-20: The authors go on about how important soil moisture is but then choose not to spin up soil moisture? This is confusing if, as the authors claim, only 10-14 days are required for spin up. Given that there was a heat wave over Europe in 2013 having the correct soil moisture field would be critical to get the proper atmospheric circulation.

Page 8 L14: "low pressure" should be replaced with "negative bias"

Page 8 L19: How is the standard deviation calculated? On mean daily values? Something else? This lack of clarity on calculations appears in other areas of the manuscript as well.

Page 8 L26: Delete "significantly". Unless describing the result of a hypothesis test this term should not be used in such a context. There are other areas of the manuscript where this is used.

Figures

As stated in the general comments the figures could benefit from inclusion of confidence bounds from the reference data.

Figure 9 can be removed, as there is no reason to expect these simulations to match the temporal march of the reanalysis.

[Figure]

---

## Author Comment (AC1) · 24 Feb 2017

We would like to thank Reviewer #1 for the very valuable comments and suggestions to improve the manuscript.

The authors have driven the atmospheric model WRF with analysis fields of the operational forecasting system of the ECMWF to simulate the period July to August 2013 in a mid-latitudinal belt (20°N to 65°N) around the globe with two grid sizes: 0.12° and 0.03°. Both simulations make use of sea surface temperatures from the Operational Sea Surface Temperature and Sea Ice Analysis (OSTIA) data. In the 0.03° simulation, parameterisation for deep convection is turned off. In order to evaluate the belt simulations, they are compared to various reference datasets (analysis fields of the operational forecasting system of the ECMWF, EOB-S, CMORPH) and in several sub-domains. In addition, the authors ask specific questions about the added value in the 0.03° simulation.

General Comments

The paper combines two relatively new innovations, convection permitting simulations and belt simulations. The strengths and weaknesses of both innovations are largely unknown and hence, the paper is worth to be published. However there are two major issues that need to be clarified first:

**Scientific quality**

The solution of a local area model is partly predominated by its lateral boundary conditions (LBCs). The larger the model domain becomes, the weaker becomes the coupling to its LBCs and the larger become large-scale deviations from its driving data in the interior of the model. Kida et al. (1991) and Paegle et al. (1996) are often cited in this context. More recently, Becker et al. (2015) demonstrated that a local area model creates artificial flows to compensate those large-scale deviations in order to achieve physical consistency with the LBCs along the lateral boundaries and that an increase of the model domain does not change this – the artificial flows simply become more complex.

In our configuration, the model is driven only by northern and southern LBCs and the SSTs. This set up makes is easier to differentiate between effects due to LBC (which should be small) and internal model physics.

In contrast, in the study of Becker et al. (2015), the secondary circulation pattern were detected in a 41 year COSMO-CLM downscaling effort using ECHAM5 simulations on a T63 grid as LBCs. Therefore, there are 2 major differences to our study: 1) we do not have a 41 year climatology, as this is simply not possible, and 2) the latitude-belt domain does not have any boundaries in west-east direction preventing deflections from the boundaries. It is also challenging to derive whether the T63 model or the high-resolution LAM is more accurate because partly the internal circulation, which was argued to be induced over the Alpine high mountain range, may be more accurate in the LAM rather than in the T63 simulations. Therefore, the results of Becker et al. (2015) are hardly applicable to our study so that we included it in our introduction as a motivation for performing latitude-belt simulations. Also in the study of Zagar et al. (2013) [Žagar, N., L. Honzak, R. Žabkar, G. Skok, J. Rakovec, and A. Ceglar (2013), Uncertainties in a regional climate model in the midlatitudes due to the nesting technique and the domain size, J. Geophys. Res. Atmos., 118, 6189–6199, doi:10.1002/jgrd.50525], the location of lateral boundaries in west-east direction disturbed the model performance. The smaller the

model domain was, the larger was the influence of the LBCs. As we do have a very large model domain where almost 50% are water surfaces, the SST forcing plays a stronger role compared to the influence of the LBCs. In order to reduce the effect of the LBCs further, we are studying here the performance of a latitude belt simulation what is to our knowledge for the first time.

The study of Kida et al. (1991) describes an alternative way to nest LAMs into coarser resolution models by applying the lateral boundaries in wavenumber space. A prerequisite is that the LAM model to be nested in the coarser model has to be a spectral model like the ALADIN and AROME models from Météo France. As WRF is not a spectral model, this method cannot be used in our case.

In the study of Paegle et al. (1996) the model was forced towards the coarser resolution simulation by applying nudging. Our intention is not to have a time-space interpolator but that the model develops its own balance for process studies and for detecting errors in model physics.

Both of these studies are not applicable, because the purpose of our study is to investigate the behavior of a latitude-belt configuration at two different resolutions which are hardly affected by LBCs in west-east direction. This type of simulation can be seen as an ensemble member of a seasonal forecast system initialized by a global model and forced by observed and simulated SSTs.

As it is well known and subject of many ongoing studies, there is predictive skill up to the seasonal scale due to the memory of the Earth system with respect to ocean circulations, soil moisture distribution, and vegetation properties. Additionally, model performance should improve on the convection permitting scale because land-atmosphere interaction is better represented particularly in heterogeneous terrain, orographic effects are simulated more accurately, and the parameterization of deep convection, which is subject of severe model errors, is turned off (Rotach et al. 2009, Wulfmeyer et al. 2011). Based on these considerations, we disagree with the reviewers that the comparison of the latitude-belt simulations with our two different resolutions cannot be compared with observations. Deviations with respect to ECMWF analyses should degrade slower and deviate with less rms in the high-resolution model even when forced only with northern and southern LBCs but with SSTs.

Therefore, we consider this study as a first steps towards the analysis of the predictive skill of seasonal ensemble members and partly of future latitude belt dynamical downscaling runs for regional climate simulations. This prospect is currently extensively discussed in the regional climate and seasonal forecast communities for the development of next generation seasonal forecast and regional climate models.

We clarified the first paragraph of the abstract and it reads now:

"Increasing computational resources and the demands of impact modelers, stake holders and society envision seasonal and climate simulations at the convection permitting resolution. So far such a resolution is only achieved with limited area model whose results are impacted by zonal and meridional boundaries. Here we present the set-up of a latitude-belt domain that reduces disturbances originating from the western and eastern boundaries and therefore allows for studying the impact of model resolution and physical parameterization. The Weather Research and Forecasting (WRF) model coupled to the

NOAH land surface model was operated during July and August 2013 at two different horizontal resolutions, namely 0.03° (HIRES) and 0.12° (LOWRES). Both simulations were forced by ECMWF operational analysis data at the northern and southern domain boundaries, and the high-resolution Operational Sea Surface Temperature and Sea Ice Analysis (OSTIA) data at the sea surface. The simulations are compared to the operational ECMWF analysis for the representation."

In the presented belt simulations, there are no western and eastern boundaries and hence, the decoupling becomes an important factor.

This is correct and the analysis of the performance of the model system under these conditions is the goal of our study.

This can be seen in principal in Fig. 9: the model creates significant anomalies in MSLP (low and high pressure systems are created that do not exist and vice versa) in the Atlantic region, but from time to time (e.g. around July 27 and August 10 to 15) the influence of the driving data becomes dominant. The fact that there is some coupling to the LBCs at all comes from the location of the sub-domain: the Atlantic region touches the northern boundary. In the interior of the model domain, the coupling might be much smaller.

We investigated a possible coupling effect from the northern boundaries and reduced the averaging domain to 60W-10W and 40N-55N (1000km away from the northern boundary), the result is very similar to that observed in Figure 10. Additionally we also investigated the 500hPa geopotential height, but again, the behavior is nearly the same independent of the selected domain across the Atlantic. Thus we are confident that the influence of the northern boundary can be neglected compared to SST forcing and internal dynamics.

A short paragraph was added to the discussion section on page 17, line 29:

"Referring to Fig. 10, the potential influence of the northern boundaries was investigated by slightly varying the domain. When selecting a much smaller domain between 60°W - 10°W and 40°N -- 55°N, the curve progression of the MSLP and 500~hPa geopotential height is very similar to the behavior shown in Fig. 10 (not shown here). This indicates that the influence of the northern boundaries on the development of the simulation compared to the SST is not significant- especially as the meridional wind speed is very weak in this area."

Hence, the simulations may be affected by large-scale decoupling to such a large extent that the entire evaluation in its current stage is flawed.
We disagree that this configuration does not allow for evaluating our model runs. The model is not only driven at the LBs but also by the SSTs. Furthermore, some predictive skill of the model is kept up to the seasonal scale. Therefore, we argue that a model with better physics and resolution will demonstrate a better performance and it should be possible to identify problems in model physics. This is an advantage of this model run.

The description of Fig. 10 now reads as follows on page 13, line 3:

"In addition, Figure 10 shows the time series of the averaged MSLP over the North Atlantic between 40°N and 65°N and 60°W and 10°E (white rectangle named Atlantic in Fig. 1. During the first ~10 days, the HIRES simulation (red line) agrees well with the ECMWF analysis while the LOWRES simulation show slightly lower pressure values. After this period, the LOWRES simulation shows considerably lower MSLP compared to the ECMWF analysis while the HIRES simulation is much closer the ECMWF analysis until day 18 of the forecast where both simulations miss the development of a depression. Both simulations

are able to capture the pressure drop after 25 days of forecast but the HIRES simulation shows a better agreement with the ECMWF analysis. In the further course, both WRF simulation overestimate the strength of the high-pressure situation with being closer to the analysis again after 45 days. Overall, the LOWRES simulation shows a tendency to even further overestimate the strength of low and high pressure systems. The mean bias of the HIRES simulation during July is 1.6 hPa while it is -0.8 hPa for the LOWRES simulation. In August, the bias of the HIRES simulation stays the same while for the LOWRES simulation it now turns into a positive bias of 2.2hPa. The root mean square error during July is 4.5 hPa and 4.65 hPa for the HIRES and LOWRES simulation, respectively. It further reduces to 3.5 hPa (HIRES) and 3.65 hPa (LOWRES) during August 2013."

Shifts in time/space between modelled and observed phenomena are limiting the applicability of traditional statistical analysis. Biases and other error measures are showing the summary effect of large-scale decoupling and model deficiencies (which should be the only focus in a model evaluation study).

We agree that this is not a traditional analysis but new model configurations need new ideas of model evaluation. We do not see it as a disadvantage that decoupling takes place but as an advantage to disentangle errors due to boundaries (strongly reduced here), model physics, and model resolution. The model is still forced by SST data and some predictive skill remains up to the seasonal scale (see above).

A common approach to overcome this mismatch is to extend the simulation period to multiple decades and evaluate statistical measures in a climatological way (as it is done for climate models, for instance). However, in the face of high computational costs, this might not be feasible.

As pointed out in the computational setup section on page 7 ff., this is currently impossible due to limited computing and storage resources. This model run was a pioneering, special project at HLRS in order to demonstrate the power of corresponding, future model configurations.

For the purpose of model evaluation it would be enough to demonstrate that the simulations are lying within the bandwidth of possible realistic developments. The climatological year-to-year variability (on a monthly basis), which could be derived from ERA-Interim, or extended ensemble forecast data (e.g. http://www.ecmwf.int/en/forecasts/datasets/set-vi) – as the forecast model runs without ingestion of observation – could be used to define such space of possible developments.

This is a great idea. We followed your suggestion and compared the 500 hPa geopotential height anomalies with the biases of the WRF forecasts. The anomalies in Figure 2 show maximum values of 60 gpm during the two month period. Allowing a factor of 2 to identify the space for possible developments and comparing the values shown in Figure 9, the HIRES simulation lies mostly within this range, while the deviations of the LOWRES simulation are larger. Figure 9 also suggests that the LOWRES simulation starts to drift away earlier from the ECMWF analysis.

According to the suggestion of Referee #2, we added an additional plot (Figure 6) showing whether both WRF simulations are within ±2 standard deviations of the operational ECMWF analysis in terms of the mean sea level pressure field and 500hPa geopotential height. Both simulations mostly stay well within ±1 standard deviations of the ECMWF operational model with advantages, especially during August, of the HIRES simulation.

The following was added to the manuscript on page 11, line 25:

"To further assess the quality of the simulation, Fig. 6 shows the confidence of the WRF simulation biases expressed in terms of ECMWF standard deviations for MSLP and 500~hPa geopotential height indicating that the bias mostly stays within ±2 standard deviations of the ECMWF analysis for both variables. The mean value of the deviation expressed in terms of ECMWF standard deviations for the MSLP is 0.22 and 0.36 (HIRES) and 0.29 and 0.43 (LOWRES) for July and August, respectively. For the deviations of the 500 hPa geopotential height the values are 0.21 and 0.24 (HIRES) and 0.21 and 0.31 (LOWRES), respectively."

Because of the large-scale decoupling that makes the simulations partly incomparable to observational or observation based data, the investigation of added value is flawed, too.

We do not agree with this statement (see above) due to remaining internal forcing by the SSTs and remaining predictive skill up to the seasonal scale.

In addition, the asked questions about added value are way too generally expressed. With one coarse and one fine resolved simulation of the same model, no robust conclusion on added value can be drawn. In such a case, the added value analysis is limited to this specific case.

We agree that based on these model runs, identification of issues with model physics are challenging. However, for the above mentioned reasons we are convinced that our analyses are still valid for the time period of the model runs and confirm an improved performance of the model running on the convection-permitting scale. The benefit is shown by a reduction of the classical scores like bias, RMSE, correlation and Pearson Skill Score of the HIRES simulation compared to the LOWRES experiment. From our results it is clear that a better representation of the terrain and land-use heterogeneity and the possibility of waiving the application of a convection parametrization leads to a better precipitation forecast as shown by the traditional scores and PSS. Most of the applied convection schemes are developed for scales of ~50km and appear do not work properly on resolution at 0.12° as also indicated in the studies of Prein et al. (2015a, Clim Dyn.) and Warrach-Sagi et al. (2013, Clim:Dyn.)

As we are able to trace model errors back to problems with model physics, the results form a basis for more detailed and more extended future studies.

Unfortunately it was not possible to perform longer term simulations. With increasing computational performance, longer simulations can be performed in the future.

By the way, for demonstrating added value, it is not enough to show that biases on a monthly basis are reduced, because monthly biases are the result of multiple processes and phenomena that may take place at the same time and also in sequences. So, a reduction of a monthly bias can be the result of enlarged process and phenomena related biases that are simply cancelling out each other. Hence, demonstrating added value includes a thorough investigation of the underlying processes and phenomena plus a demonstration that these processes and phenomena are more properly captured by the finer resolved model.

While admitting that the evaluation of the models is difficult, a reduction of bias during the simulated time period is important and significant.

When increasing the horizontal resolution of the model, several ambient conditions are improved: 1) the representation of the terrain is much more realistic as compared to the 12 km run. The publication of Prein et al (2015), which was cited by you below, points towards that a resolution increase from 0.44 to 0.11 degree still suffers from the windward-lee effect. Only a further increase to the CP scale gives a chance to considerably improve precipitation (e.g. Prein et al, 2015, Rev. Geophys) 2) The land-use cover, soil texture and its variability is also much more realistically represented on the higher resolution which is absolutely necessary 3) The high-resolution SST combined with better resolved coast lines will improve coastal effects.

As the RMSE (PSS) of mean sea level pressure, geopotential height and especially precipitation are considerably reduced (increased), we are convinced that atmospheric processes and phenomena are more properly captured by the simulation on a convection permitting resolution.

Prein, A. F., W. Langhans, G. Fosser, A. Ferrone, N. Ban, K. Goergen, M. Keller, M. Tölle, O. Gutjahr, F. Feser, et al. (2015), A review on regional convection-permitting climate modeling: Demonstrations, prospects, and challenges, Rev. Geophys., 53, 323–361. doi:10.1002/2014RG000475.

To solve this issue, the authors could either include such a thorough process and phenomena based analysis or should put more effort on the model evaluation and its problematic (see above) and do not announce added value in such a prominent way.

We agree that thorough process and phenomena based analysis would strengthen the evaluation of our simulations.

The difficulty is to obtain suitable observations for the whole model domain which allow a fair comparison of e.g. diurnal cycles of temperature, wind, and precipitation. At first glance, the ECMWF data availability chart for conventional observations (see e.g. http://www.ecmwf.int/en/forecasts/charts/monitoring/dcover?time=2017020100,0,2017020100&obs=synop-ship) shows a nice coverage, but a closer inspection reveals that most of the stations only report in 3h or 6h intervals. If considering wind and (hourly) precipitation observations, the station density dramatically reduces. From the ECMWF analysis or ERA-Interim data, no diurnal cycles can be displayed since only 6 hourly data are available. Therefore, we would like to keep this suggestion for future studies when suitable model and observational data sets are available.

Since large-scale decoupling plays such an important role, it needs to be an integrative part of the discussion (Section 5) and summary (Section 6).

Yes we agree with your suggestion. The discussion section was reordered and the following was added to address the decoupling on page 16, line 13:

"As the simulations are only driven by high-resolution SST data and no zonal lateral boundaries are applied, this can be isolated to the applied model configurations. The Pacific and North Atlantic are the most sensitive areas with respect to the development of storms (Fig. 7a,b), thus small differences in temperatures due to the applied model physics can lead to different spatial and temporal evolutions of storm systems."

The summary was also reordered and now contains information on the decoupling starting on page 19, line 13.

**Presentation quality**

Reference data, error measures, including an explanation why theses reference datasets and error measures are selected and how data from different grids is remapped onto a common evaluation grid is missing in the experimental setup (Section 2). Instead this information is (partly) given at other places, for instance at the beginning of the result section (Section 4). Having an evaluation concept in section 2 summarising all of this would increase the readability of the manuscript.

We agree. The paragraph dealing with the observational data set was moved to the experimental setup section on page starting now on page 6, line 33.

Specific Comments

The following study would nicely fit into the introduction section:

Prein, A. F., A. Gobiet, H. Truhetz, K. Keuler, K. Goergen, C. Teichmann, C. Fox Maule, E. van Meijgaard, M. Déqué, G. Nikulin, R. Vautard, A. Colette, E. Kjellström, and D. Jacob (2015), Precipitation in the EURO-CORDEX 0.11˚ and 0.44˚ simulations: High resolution, high benefits?, Climate Dynamics, 46, 383–412, 10.1007/s00382-015- 2589-y

This paper was added in the introduction as a further motivation to go to the convection-permitting scale because in this work the remaining deficiencies of models running on grids with convection parameterizations are suffering from severe errors such as the windward-lee effect in orographic terrain (Wulfmeyer et al. 2011) making their input almost useless for most end users such as hydrologists. This reference was added to the third paragraph of the introduction on page 2, line 28. We also added the reference of Prein et al. (2015, Rev. Geophys) to the introduction on page 3 line 10 being in favor for the necessity of convection permitting scale simulations.

Page 3, lines 14, 15: It is unclear how errors in the large scale circulation patterns can be traced back to the applied physics schemes, especially in the light of large-scale decoupling.

As we did not apply zonal LBCs, the main atmospheric flow in this direction is not disturbed by different physics between the LAM. Thus, deviations in the atmospheric flow can be related to the applied model as especially the CP resolution is much closer to reality in terms of terrain, coast lines, and land use than the driving model.

We corrected the sentence on page 3, line 35 and it now reads: "As the general circulation is west-east oriented and lateral forcing is only applied at the northern and southern boundaries, e.g. errors in the large scale circulation patterns can be traced back to the applied model with its specific physics schemes. The model physics of the coarser resolution model (ECMWF) providing the lateral boundaries in south-north direction only plays a minor role."

Page 3, lines 22, 23: It is not clear how large uncertainties over the Atlantic and Pacific can be explained by differences in model physics, especially in the light of large-scale decoupling which is also active when model physics are identical (simply extending the model domain

of a local area model by some grid cells into one direction gives different results; see Becker et al., 2015).

Both areas are the most active regions in terms of the Jet stream and tropical storms. Thus small differences e.g. in the temperature fields between WRF and ECMWF can lead to an amplification of the development of weather systems potentially leading to phase shifts and different storm tracks. In case a classic LAM approach is used, the internal variability of the nested model is strongly influenced by the boundary conditions. This is not the case if a latitude-belt is applied.

The paragraph on page 4, line 9 was modified and it reads now:

"Their results indicate a strong influence of the zonal LBCs on the internal model variability due to different model physics and the applied nesting technique. In case the model domain is made smaller and smaller, the RCM does not have the chance to develop its own internal variability and the results are mainly driven by the LBCs. This means that the analysis of the model errors is giving more insights into the applied model in case of a latitude-belt set-up."

Page 5, line 14: What is the advantage of using OSTIA instead of SST from the operational ECMWF analyses? OSTIA is given on a daily resolution and need to be interpolated in time (see page 5 line 34 to page 6, line 3), while SST from ECMWF is already on a 6 h basis. I am not an SST expert, but a short literature research brought up a paper from Seo et al. (2014) which demonstrates the importance of sub-daily SST variability to properly capture the onset and intensity of Madden–Julian oscillation (MJO) convection in the Indian Ocean in a coupled WRF-ocean model. Seo, H., A. C. Subramanian, A. J. Miller, and N. R. Cavanaugh (2014), Coupled Impacts of the Diurnal Cycle of Sea Surface Temperature on the Madden–Julian Oscillation. J. Climate, 27, 8422–8443, doi: 10.1175/JCLI-D-14-00141.1.

The major advantage of OSTIA is the native resolution of 1/20° (5km) while the SST data from the operational analysis would be on the same resolution of as the driving data (0.125° in our case). The higher resolution of the SST data becomes especially important in coastal regions (e.g. Himada, S., Ohsawa, T., Kogaki, T., Steinfeld, G., and Heinemann, D. (2015), Effects of sea surface temperature accuracy on offshore wind resource assessment using a mesoscale model. Wind Energy, 18, 1839–1854. doi: 10.1002/we.1796).

The ECMWF applies the daily OSTIA SST data set at initial time and the SST is kept constant during the operational 10-day forecast (section 8.9 of http://www.ecmwf.int/sites/default/files/elibrary/2013/9245-part-iv-physical-processes.pdf). This also means that the SST data from ECMWF are constant throughout the day (see chapter 12 in http://www.ecmwf.int/sites/default/files/elibrary/2013/9243-part-ii-data-assimilation.pdf).

The study of Seo et al. is very interesting, however our study region covers only a small part of the tropics and thus we assume is it feasible to use constant SST data throughout the day. Also this study applies a very coarse resolution together with the necessary convection parametrization which is well known to deteriorate the quality of precipitation forecasts.

The paragraph on page 6, starting line 17 now reads:

"Forcing data at the northern and southern boundaries were provided by 6 hourly ECMWF operational analysis data on model 15 levels and are blended with the default linear decay over five grid points into the WRF model grid. Sea Surface temperatures were provided by

the high-resolution Operational Sea Surface Temperature and Sea Ice Analyis (OSTIA) data (Donlon et al., 2012) with a resolution of  5km. As this study only contains a small part of the tropics, it appears practicable to use more or less constant SST data for each day. As they are only available in daily intervals, these data were linearly interpolated to the 6 h intervals of the ECMWF analysis. This interpolation was performed by using version 1.7.0 of the Climate Data Operators 20 (CDO;https://code.zmaw.de/projects/cdo)."

Page 6, lines 11 to 17: the 0.03° and 0.12° simulations make use of pnetcdf. A discussion about pnetcdf is missing here. Does pnetcdf solve the problem?

PNetCDF only reduces the amount of computing time that is spend for I/O. The model results are the same no matter which version of NetCDF is applied. The I/O rates on the system used for this study is around 7GB/s with PNetCDF while it is only around 500MB/s with serial NetCDF. The CDF5 format convection is only available when using PNetCDF. Serial NetCDF does not offer this capability.

The paragraph on page 9, line 5 was enhanced and now reads:

"If even a higher number of grid points is planned to use, one has to take care about the NetCDF limitations in the presently used CDF-2 format. This convention only allows $2^{32}-4$ bytes per array which can be too small for future experiments so that the new CDF-5 standard has to be considered. This feature is available from PNetCDF version 1.6.0 onwards.

In order to have the possibility to apply such a large domain latitude-belt simulation on the CP scale, we modified the source code by exchanging the second argument of the nf_create function from NF_64BIT_OFFSET to NF_64BIT_DATA in frame/module_bdywrite.F. A similar change was performed in external/io_pnetcdf/wrf_io.F90  In the NFMPI_CREATE  function, the third argument has to be replaced by NF_64BIT_DATA. In the NFMPI_OPEN  function, NF_NOWRITEhas to be replaced by NF_WRITE."

Page 8, line 25: What is "good agreement"? What biases are acceptable? (These questions should be tackled in an evaluation concept in section 2.)

We agree that a more precise definition is necessary. Following the study of Kotlarski et al, who evaluated a 20-year forecasting ensemble, a mean sea level pressure bias of 3 hPa is acceptable. For temperature, mean deviations of 3 °C are tolerable in homogenous terrain while for precipitation a relative difference of 100 % is acceptable. Where precipitation amounts are low like in Africa, even a difference of more than 100% is acceptable.

We added this short paragraph to the verification data strategy section on page 7, line 18.

"Following the study of Kotlarski et al. (2014), who evaluated a 20-year forecast ensemble, a mean sea level pressure bias of 15 3 hPa is acceptable. For temperature, mean deviations of up to 3°Care tolerable in homogeneous terrain while for precipitation relative differences of 100% are reasonable. In case of very low precipitation amounts like in North Africa, relative deviations of more than 100% are tolerable."

Kotlarski, S., Keuler, K., Christensen, O. B., Colette, A., Déqué, M., Gobiet, A., Goergen, K., Jacob, D., Lüthi, D., van Meijgaard, E., Nikulin, G., Schär, C., Teichmann, C., Vautard, R., Warrach-Sagi, K., and Wulfmeyer, V.: Regional climate modeling on European scales: a joint standard evaluation of the EURO-CORDEX RCM ensemble, Geosci. Model Dev., 7, 1297-1333, doi:10.5194/gmd-7-1297-2014, 2014.

Page 11, lines 7 to 8: There must be something fundamentally going wrong with the model in this specific region. Maybe it is related to the initialisation of the soil.

We investigated the soil moisture content in this area. The ECMWF analysis fields and the temporal evolution of the WRF soil moisture data is very similar.

The authors are encouraged to contact WRF experts (e.g. Walter Immerzeel, University of Utrecht, or the CORDEX-South-Asia or CORDEX-Central-Asia communities) that are operating the model in this region. There is also a new reference dataset for temperature available. It is called WFDEI (Weedon et al., 2010; 2011) and can be downloaded from ftp://rfdata:forceDATA@ftp.iiasa.ac.at

Weedon, G.P., Gomes, S., Viterbo, P., Österle, H., Adam, J.C., Bellouin, N., Boucher, O., and Best, M., 2010. The WATCH Forcing
Data 1958-2001: a meteorological forcing dataset for land surface- and hydrological models. WATCH Tech. Rep. 22, 41p (available at www.eu-watch.org/publications ).

Weedon, G.P., Gomes, S., Viterbo, P., Shuttleworth, W.J., Blyth, E., Österle, H., Adam, J.C., Bellouin, N, Boucher, O., and Best, M., 2011. Creation of the WATCH Forcing data and its use to assess global and regional reference crop evaporation over land during the twentieth century. J. Hydrometerol. 12, 823-848, doi: 10.1175/2011JHM1369.1.

This dataset is based on ERA-Interim, but it corrects temperatures in a way that it is consistent with the observed behaviours of glaciers in the Himalayan region. It might be a more reliable reference dataset than the ECMWF analysis fields.

The WATCH data set is based on ERA-40 downscaled to a 0.5° grid to match the CRU land mask. We compare our model with the ECMWF operational analysis which already considers a lot of 2-m temperatures which are used in the 4DVAR analysis (see e.g. http://www.ecmwf.int/en/forecasts/charts/monitoring/dcover?time=2016122200,0,201612 2200&obs=synop-ship). To complement 2-m observations, ECMWF also applies assimilation of satellite based surface temperature observations. At this time, the 4DVAR was performed on an outer loop T799 grid with 137 levels (cycle 38r2) while the ERA-Interim analysis is performed on a T255 grid with only 62 levels. Assuming a better resolved terrain and underlying land-use data set in the operational HTESSEL land-surface model, the operational analysis should be superior to even a corrected ERA-40 analysis on a 0.5° grid.

The large temperature deviations over the Steppe regions are also observed in a study of Zhang et al. (2014, J. Hydromet). They conclude that the soil hydraulic parameters used in the NOAH LSM are inappropriate in steppe regions. Currently these variables are read in from tables so it is difficult to adjust these values for specific regions in case of a large model domain covering different climate regimes. The regions showing large biases of the 2-m temperatures (California, eastern Canada, and China/Mongolia) also exhibit large deviations of the upper soil temperatures compared to the ECMWF analysis fields.. As a 4DVAR also includes requires a forecast model, in this case the ECMWF operational model with different physics compared to WRF, this can also lead to differences between the WRF simulations and the analysis in case the observations density is low.

Another possible factor are the values for the background albedo. ECMWF uses a climatological value while the WRF model offers monthly varying albedos. A closer look into both fields revealed that in these specific areas, the ECMWF albedo values are

higher by about 5-10% leading to ~50-100W/m² more solar radiation absorbed by the ground.

A reference to the study of Zhang et al. is given in the discussion on page 18, line 18.

"As pointed out by Zhang et al. (2014), the soil hydraulic parameters used in the NOAH LSM show some deficiencies in desert and steppe regions over Inner Mongolia. As our study already makes use of an improved version of the thermal roughness length calculation over land (Chen and Zhang, 2009), it appears that a more proper description of the canopy resistance over the desert steppe can be beneficial. At present, the WRF model system unfortunately does not offer the possibility of latitude or region varying parameters for the land-surface models."

Zhang, G., G. Zhou, F. Chen, M. Barlage, and L. Xue, 2014: A Trial to Improve Surface Heat Exchange Simulation through Sensitivity Experiments over a Desert Steppe Site. J. Hydrometeor., **15**, 664–684, doi: 10.1175/JHM-D-13-0113.1.

Technical Corrections

Page 3, line 16: Is there a reference for the storm systems affecting Europe?

With the citation of Rogers (1997) we added a reference to the storm track climatology over the Atlantic Ocean on page 4, line 3.

Page 3, line 33: typo – "This study is organised . . ."

This was corrected on page 4, line 1314.

Page 4, line 14: Euro-CORDEX should be referenced by Jacob et al. (2014).

Jacob, D., J. Petersen, B. Eggert, A. Alias, O. B. Christensen, L. M. Bouwer, A. Braun, A. Colette, M. Déqué, G. Georgievski, E. Georgopoulou, A. Gobiet, L. Menut, G. Nikulin, A. Haensler, N. Hempelmann, C. Jones, K. Keuler, S. Kovats, N. Kröner, S. Kotlarski, A. Kriegsmann, E. Martin, E. van Meijgaard, C. Moseley, S. Pfeifer, S. Preuschmann, C. Radermacher, K. Radtke, D. Rechid, M. Rounsevell, P. Samuels- son, S. Somot, J.-F. Soussana, C. Teichmann, R. Valentini, R. Vautard, B. Weber, and P. Yiou (2014), EURO-CORDEX: New high-resolution climate change projections for European impact research, Regional Environmental Change, 14, 563–578, 10.1007/s10113-013-0499-2.

This reference is added to the introduction on page 5, line 14.

Page 5, line 18: Are there any references for the studies that have shown the spin-up time for NOAH's land surface model?

E.g. Angevine et al. 2014 [Angevine, W. M., Bazile, E., Legain, D., and Pino, D.: Land surface spinup for episodic modeling, Atmos. Chem. Phys., 14, 8165-8172, doi:10.5194/acp-14-8165-2014, 2014.] performed spin-up experiments using the NOAH LSM. In their study, WRF was driven with coarse resolution ERA-Interim data. Their study indicates that already after one day, the spatial structure of the soil moisture is clearly visible.

We also investigated the soil moisture over Europe. Only minor differences are visible in the first three layers for about 3 weeks between WRF and ECMWF. At the end of July, the WRF soil moisture starts to deviate from ECMWF, probably as a result of different precipitation amounts. The soil moisture in the 4th layer is very homogenous although the WRF model shows lower values compared to ECMWF. The 4th layer of the ECMWF analysis covers the depth between 100cm and 255cm, while the 4th layer of the NOAH

model covers the depth between 100cm and 200cm. Therefore the observed slow decay of the ECMWF soil moisture can be explained.

The following was added to the manuscript on page 6, line 24:

"Soil moisture and temperature were initialized from the ECMWF operational analysis. The Hydrology land-surface model HTESSEL (Balsamo et al., 2009) assimilates ASCAT soil moisture data since 2012 (Albergel et al., 2012). A brief comparison of the analyzed ECMWF soil moisture and HIRES soil moisture data over Europe revealed no major differences between both data sets during the first 17 forecast days. The absolute soil moisture content in the three topmost layers is between 0.25 and 0.3 m³/m³ and the differences between HIRES and ECMWF vary around 0.05 m³/m³. This is very promising especially as ECMWF assimilates ASCAT soil moisture data since 2012 (Albergel et al., 2012). Thus it appears feasible to waive a separate spin-up run for this two month period. Afterwards, the soil moisture shows a different behavior most probably due to different evapotranspiration and precipitation patterns."

Page 9, line 1 to 2: Is there a reference to the analysis on tropical storms of JMA?

A link to the JMA website (http://www.jma.go.jp/en/typh/ ) was added on page 12, line 13.

Page 9, line 5: Is there a reference to this jet stream north of the Tibetan Plateau which is typical for the summer monsoon?

A reference to Xie et al. (2015, Journal of Climate) was added on page 12, line 16.

Page 9, line 10: up to now, it was not clear that RMSE is used at all. It is also not clear which RMSE is used (the RMSE of monthly means or on a daily basis or whatsoever)

Thanks for the comment. As we do not show the RMSE for the sake of brevity, we clarified the sentence that the RMSE is not shown here.

RMSE and standard deviation are calculated on a daily basis. This was clarified in the Verification data strategy section on page 7, line 13. "Standard deviation, bias and RMSE are calculated on a daily basis by comparing the 12Z time steps for each day. The scores are finally averaged over the two month period."

For the verification of precipitation, we also added a sentence on page 8, line 2:

"The RMSE and biases with respect to the CMORPH and E-OBS data sets are calculated from the two month accumulated precipitation over the whole observation domain."

Page 10, line 8: typo – ". . . due to small . . ."

This typo was corrected.

Page 10, line 19: typo – ". . . caused by the inaccurate . . ."

This typo was corrected on page 14, line 1

Page 15, line 29 to 30: cumulus parameterisation is also changed in LOWRES. Is GRIMS also active in LOWRES?

Yes, GRIMS is also active in the LOWRES simulation. We ensured that the description of the model setup on page 5, line 32 is clear on that point.

Figure 2: colours of the shades are too intensive, continents can only hardly be seen. Also the structure of the plot should be consistent with Figure 4 (anomalies should be shaded and the climatology should be in solid lines).

The shaded fields and contour lines were swapped so that the anomalies are shaded and the climatology is shown by contour lines. Also the color of the geophysical border in Figures 2 and 4 has been changed to blue for a better distinction.

[revised manuscript text omitted]

---

## Author Comment (AC2) · 24 Feb 2017

We would like to thank Reviewer #2 for the valuable comments and suggestions to improve the manuscript.

The authors investigate the benefits of convection permitting modeling by employing the WRF modeling system in a channel configuration over the Northern Hemisphere (between 20° N and 65°N) at resolutions of 0.12° (LOWRES) and 0.03° (HIRES), respectively. In HIRES the deep convection parameterization scheme is turned off. The necessarily short integration period covers the summer of 2013 (July and August). This period was notable for exhibiting a strongly positive phase of the North Atlantic Oscillation and generally weaker subtropical highs over the Atlantic and Pacific basins. The driving data is from the ECMWF operational analysis data at the northern and south- ern boundaries and the OSTIA 5km SST data set at the sea surface. The authors then compare their results to reanalysis and gridded combined data products (e.g., E-OBS,CMORPH). They aim to answer the three questions: 1) What is the benefit of a CP resolution with respect to the spatial representation of large-scale features in comparison to coarse resolution? 2) Does the higher resolution lead to an improvement of surface variables such as 10m windspeed and 2m temperatures? 3) What is the benefit of the CP resolution with respect to the spatial distribution and amount of precipitation?

Given large channel domain and the fact that convective permitting simulations are still relatively rare this study can potentially make a useful contribution to our understanding of how and why simulations at these grid spacings are useful. However, the experiment is not designed in such a way that it can answer the questions as they are posed. The shortcomings are detailed below as are suggests for improvement. I will focus on what I see as the two major issues that must be addressed before the manuscript can move forward. There are likely more specific comments but these can be addressed in the next revision.

**General comments**

Due to the channel set up the model simulations are largely "free".

This is not entirely correct. The simulations are still driven by observed SST data and thus the simulation is only partly free. The relative area covered by the SSTs is increasing with the latitude belt configuration in contrast to an LAM; therefore, increasing the domain and removing the west-east boundaries has also considerable advantages with respect to model performance.

In other words internal variability can account for much of the difference that we see between the simulations and the reference data sets. In fact there is little reason to assume that they would in anyway resemble each other. In the absence of nudging the one-to-one comparison of the model simulations with the reference fields for the large-scale circulation is doomed to fail.

Of course one cannot expect that the model reproduces the observation (ECMWF analysis in this case) one-to-one since in contrast to WRF, ECMWF uses sophisticated data assimilation of a huge set of observations to keep the analysis close to the real observations. However, comparing these simulations with high-resolution (re-)analysis and observational data is the only way to specify how the model is performing on different resolutions and different physics.

As both Reviewers address a similar point here, we are adding the response to reviewer 1:

The purpose of our study is to investigate the behavior of a latitude-belt configuration at two different resolutions which are affected by LBCs in west-east direction. This type of simulation can be seen as an ensemble member of a seasonal forecast system initialized by a global model and forced by observed and simulated SSTs.

As it is well known and subject of many ongoing studies, there is predictive skill up to the seasonal scale due to the memory of the Earth system with respect to ocean circulations, soil moisture distribution, and vegetation properties. Additionally, model performance should improve on the convection permitting scale because land-atmosphere interaction is better represented particularly in heterogeneous terrain, orographic effects are simulated more acccurately, and the parameterization of deep convection, which is subject of severe model errors, is turned off (Rotach et al. 2009, Wulfmeyer et al. 2011).

Based on these considerations, we disagree with the reviewers that the comparison of the latitude-belt simulations with our two different resolutions cannot be compared with observations or analyses. Deviations with respect to ECMWF analyses should degrade slower and deviate with less rms in the high-resolution model even when forced only with northern and southern EBCs but with SSTs.

Therefore, we consider this study as a first steps towards the analysis of the predictive skill of seasonal ensemble members and partly of future latitude belt dynamical downscaling runs for regional climate simulations. This prospect is currently extensively discussed in the regional climate and seasonal forecast communities for the development of next generation seasonal forecast and regional climate models.

We clarified this in the first paragraph of the abstract and it reads now:

"Increasing computational resources and the demands of impact modelers, stake holders and society envision seasonal and climate simulations at the convection permitting resolution. So far such a resolution is only achieved with limited area model whose results are impacted by zonal and meridional boundaries. Here we present the set-up of a latitude-belt domain reduces disturbances originating from the western and eastern boundaries and therefore allows for studying the impact of model resolution and physical parameterization. The Weather Research and Forecasting (WRF) model coupled to the NOAH land surface model was operated during July and August 2013 at two different horizontal resolutions, namely 0.03° (HIRES) and
0.12° (LOWRES). Both simulations were forced by ECMWF operational analysis data at the northern and southern domain boundaries, and the high-resolution Operational Sea Surface Temperature and Sea Ice Analysis (OSTIA) data at the sea surface. The simulations are compared to the operational ECMWF analysis for the representation."
.

This was also clarified in introduction on page 4, line 15:

"The goal of this study is to evaluate the benefit of a unique convection permitting latitude-belt simulation over a 2 month period which is not disturbed by zonal lateral boundaries. Due to its high computational demand it is investigated in comparison with a lower resolution set-up as currently applied in seasonal global forecasts. The simulations were not performed in forecast mode and without data assimilation as the study is considered as a pilot study for future convection permitting seasonal forecasting. In this study, we would like to answer whether a convection permitting latitude-belt simulation for a two month period improves the model performance compared to a commonly applied coarser resolution of 0.12°."

This is illustrated most clearly through examination of the Figures 4 and 5. The dominant anomalies in the large-scale circulation for 2013 are a weakening of the subtropical highs over the ocean basins and a strengthening of the low-pressure anomalies over the Eurasia. The so-called model biases wipe out this weakening over the subtropical highs and intensify the low-pressure anomalies over the Eurasia. From there the rest of the comparisons are uninformative at best.

Based on the arguments above, we disagree with this particular statement and hope that these are convincing for the reviewers.

Therefore, I would suggest the authors focus on whether this type of simulation is fit for purpose.

Yes, absolutely, the simulation fits our purposes and other research teams working on regional climate and seasonal simulations. Zagar et al. (2013) showed that large disturbances can occur in case of classical LAM simulations, which is still to date a great matter of concern. Our configuration can be the starting point for reducing basic problems of LAMs.

Žagar, N., L. Honzak, R. Žabkar, G. Skok, J. Rakovec, and A. Ceglar (2013), Uncertainties in a regional climate model in the midlatitudes due to the nesting technique and the domain size, J. Geophys. Res. Atmos., 118, 6189–6199, doi:10.1002/jgrd.50525.

In other words, can the model perform the task for which it is intended and does the HIRES simulation perform this task better, or more accurately than the LOWRES simulation?

See above.

One way the authors could do this would be to show, via hatching for example, areas where the modeled field falls outside the +/- 2 standard deviation confidence bounds of the observations. Given that these simulations are basically single realizations of internal variability, weakly constrained by the lower and north/south boundaries this is a more fair and appropriate comparison.

Thank you for your suggestion. A new Fig. 6 was included indicating that the mean sea level pressure bias stays well within 2 standard deviation of the ECMWF analysis. Additionally, the same type of plots for the 500 hPa geopotential height has been included. Here, the beneficial influence of the higher resolution is clearly visible especially during August.

The following was added to the manuscript on page 11, line 25:

"To further assess the quality of the simulation, Fig. 6 shows the confidence of the WRF simulation biases expressed in terms of ECMWF standard deviations for MSLP and 500~hPa geopotential height indicating that the bias mostly stays within ±2 standard deviations of the ECMWF analysis for both variables. The mean value of the deviation expressed in terms of ECNWF standard deviations for the MSLP is 0.22 and 0.36 (HIRES) and 0.29 and 0.43 (LOWRES) for July and August, respectively. For the deviations of the 500~hPa geopotential height the values are 0.21 and 0.24 (HIRES) and 0.21 and 0.31 (LOWRES), respectively."

Another solution would be to re-run the experiments, but constrain the flow so that expectation could be that the model, in the absence of internal model errors, would reproduce the temporal evolution of the weather over the course of July and August 2013.

Re-running these experiments is simply not possible as this implies blocking half of the current HPC system for almost three days to finish in reasonable time including post processing all data. Also disk space is an issue and is currently not available on the system.

The other issue relates to added value. Given the issues described above and the face that this is a single model, case study experiment, it is very, very, difficult to convincingly argue for added value. Rather than focus on added value using such measure as Taylor diagrams and RMSE, the authors could perhaps focus more on processes that are more accurately captured in the HIRES simulation. Examples are diurnal cycles of winds and precipitation, blocking associated with heat waves, etc.

In general, it is a good idea to compare diurnal cycles. The difficulty is to obtain suitable observations for the whole model domain which allow a fair comparison. It is especially difficult to obtain observations for wind measurements and hourly precipitation. At first glance, the ECMWF data availability chart for conventional observations (see e.g. http://www.ecmwf.int/en/forecasts/charts/monitoring/dcover?time=2017020100,0,201702010 0&obs=synop-ship) shows a nice coverage, but a closer inspection reveals that most of the stations only report in 3h or 6h intervals. If considering wind and (hourly) precipitation observations, the station density dramatically reduces. From the ECMWF analysis or ERA-Interim data, no diurnal cycles can be displayed since only 6 hourly data are available. Therefore, we would like to keep this suggestion for future studies when suitable model and observational data sets are available.

The authors should not focus on spatial comparisons as there is little reason to expect high spatial correlation between the simulations and the reference data other than that due to the fact there are climatological patterns that the simulations will somewhat follow. If the authors are really set on showing added value then I would recommend they use something like the Perkins skill score which assesses the similarity of two pdfs (Perkins et al. 2007). This metric is quite a bit more informative than the approaches shown used in the manuscript, which rely heavily on visual inspection.

We do not claim that a high spatial correlation can be expected. Nevertheless, the model should be able to remain close to large scale patterns for the reasons mentioned above. Therefore we consider it as essential to include spatial comparisons here, which also helps to detect model errors.

We followed your suggestion and calculated the Perkins skill score in addition to the other error measures. The result for the averaged August MSLP reveals a value of 0.86 for the LOWRES and 0.91 for the HIRES simulation. For the precipitation over Europe during the whole period, the Perkins score is 0.75 for the LOWRES and 0.84 for the HIRES simulation.

The following sentences were added to the results section on page 11, line 13:

"The PSS of the MSLP over Europe for the HIRES simulation is 0.91 for July and 0.9 for August 2013 whereas the LOWRES simulation yields values of 0.92 and 0.86 for July and August, respectively. Perkins et al. (2007) and Devis et al. (2013) suggest that a PSS of 0.7

indicates a reasonable model performance when compared to reference data sets. Therefore the achieved scores indicate a good performance of both WRF simulations over Europe with better results in the HIRES simulation on longer time scales."

The following sentence was added on page 15, line 28:

"The PSS during the two month period yields a value of 0.75 for the LOWRES simulation and a value of 0.84 for the HIRES simulation."

We also added a short paragraph on page 11, line 14.:

"Perkins et al. (2007) and Devis et al. (2013) suggest that a PSS of 0.7 indicates a reasonable model performance when compared to reference data sets. Therefore the achieved scores indicate a good performance of both WRF simulations over Europe with better results in the HIRES simulation on longer forecast lead times."

Devis, A., N. P. M. van Lipzig, and M. Demuzere (2013), A new statistical approach to downscale wind speed distributions at a site in northern Europe, J. Geophys. Res. Atmos., 118, 2272–2283, doi:10.1002/jgrd.50245.

Perkins, S. E., Pitman, A. J., Holbrook, N. J., & McAneney, J. (2007). Evaluation of the AR4 climate models' simulated daily maximum temperature, minimum temperature, and precipitation over Australia using probability density functions. Journal of climate, 20(17), 4356-4376.

**Specific comments**

The abstract is much too long and without critical insight. The abstract should not just be a laundry list of the results but a brief exposition of key findings. The reader should immediately grasp why this paper is of interest. The contribution this study is making should come through in the abstract.

The abstract was modified so that the intention of our study and the main results of our study are more clearly visible.

Page 5 L3-20: The authors go on about how important soil moisture is but then choose not to spin up soil moisture? This is confusing if, as the authors claim, only 10-14 days are required for spin up. Given that there was a heat wave over Europe in 2013 having the correct soil moisture field would be critical to get the proper atmospheric circulation.

We agree that this is an important topics. With this large domain, we start with an ECMWF soil moisture analysis. While admitting that a spin up run would have been better we found in many of our studies that due to a similar physics, ECMWF and NOAH soil moisture is quite similar and that the required spin up time is short.

The following was added to the manuscript on page 6, line 24:

"Soil moisture and temperature were initialized from the ECMWF operational analysis. The Hydrology land-surface model HTESSEL (Balsamo et al., 2009) assimilates ASCAT soil moisture data since 2012 (Albergel et al., 2012). A brief comparison of the analyzed ECMWF soil moisture and HIRES soil moisture data over Europe revealed no major

differences between both data sets during the first 17 forecast days. The absolute soil moisture content in the three topmost layers is between 0.25 and 0.3 m³/m³ and the differences between HIRES and ECMWF vary around 0.05 m³/m³. This is very promising especially as ECMWF assimilates ASCAT soil moisture data since 2012 (Albergel et al., 2012). Thus it appears feasible to waive a separate spin-up run for this two month period. Afterwards, the soil moisture shows a different behavior most probably due to different evapotranspiration and precipitation patterns."

Page 8 L14: "low pressure" should be replaced with "negative bias"

This was replaced.

Page 8 L19: How is the standard deviation calculated? On mean daily values? Something else? This lack of clarity on calculations appears in other areas of the manuscript as well.

Standard deviation and RMSE values were calculated on a daily basis at the 12Z time steps and finally averaged over the month. This was clarified in the Verification data strategy now starting on page 6.

Page 8 L26: Delete "significantly". Unless describing the result of a hypothesis test this term should not be used in such a context. There are other areas of the manuscript where this is used.

As we did not perform any statistical significance test, this word is replaced by "considerably".

Figures

As stated in the general comments the figures could benefit from inclusion of confidence bounds from the reference data.

We included the new Figure 6 to display confidence bounds with respect to the reference data of MSLP and 500 hPa geopotential height.

Figure 9 can be removed, as there is no reason to expect these simulations to match the temporal march of the reanalysis.

Due to the inclusion of the new Figure 6, the Figure you refer to is now Figure 10. We decided to keep Figure 10 as this figure nicely shows that the general trend of the HIRES simulation is in a better agreement with ECMWF compared to the LOWRES simulation.

[revised manuscript text omitted]

---

## Author Response (AR2)

We would like to thank Reviewer #1 again for the valuable suggestions.

General Comments

The paper has significantly improved. In particular, the comparison with the two-sigma ranges from the ECMWF analysis fields demonstrates that (1) the authors have successfully applied the WRF model in the belt set up (also in a convection permitting configuration) and (2) the model gives reasonable results that are lying within a realistic bandwidth most of the time (when the model exceeds this bandwidth the authors try to give reasonable explanations, like for the Western Pacific region). However, this does not solve the concerns about the large-scale decoupling/internal variability, but since the presented work is introduced as a pilot study and since the overall afford for doing thorough investigations based on long term evaluation simulations is way too high, the presented evaluation can be seen as a successfully passed plausibility check.

We appreciate these considerations of the reviewer concerning our work.

The authors are encouraged to provide a more complete picture of the belt configuration and to discus its strengths and weaknesses in a direct manner. Especially, its relationship to internal model variability and its consequences for the model's comparability with observational data in the face of short-term (2 months) simulations should be addressed. This does not belittle the authors' work, but would demonstrate their attempt in taking care of their article to avoid potentially misguided interpretations. A clear and direct presentation of the facts would furtherly increase the quality of the paper.

We agree that this is an important goal and are happy to refine the work as long as it is possible at this stage considering the limited time duration of the model runs and the observational data base.

Specific comments

Page 3 line 3-4: The secondary circulations described in Becker et al. (2015) are the result of lateral boundaries in general, not only in west-east direction. The manuscript is very suggestive at that point. Especially, since Zagar et al. (2013) have clearly demonstrated the sensitivity of the model results with respect to the meridional extension of the belt (see their figure 4 and figure 5).

We reformulated the sentence to avoid any ambiguities. It now reads on page 3, lines 3-7:

"The secondary circulations appear to be the result of necessary domain boundaries when downscaling a GCM. However, these deviations should be reduced in our case, as the west-east boundaries are not present anymore. The study of remaining effects in dependence of limited-area or latitude belt configurations can hardly be studied within our work due to the limited duration of the model runs so that these need to be kept for future studies."

However, it is very difficult to disentangle these results in our model simulations due to the short duration of our run. Furthermore, we still suspect that the deviations between the global model and the limited-area model runs in Becker et al. (2015) may not be entirely an error of the limited-area model, as the high-resolution model better resolves orographic effects in the central domain which are barely represented on a 1.875° model grid. In contrast to our study, the results were not compared with observations.

Page 6, line 6: As a consequence of Reynolds averaging which is the basis for a discrete differences scheme, model output on single grid cells can only be interpreted as grid cell averages. Mapping one model grid onto another by means of a bilinear interpolation associates the centre of the grid cells with their values. This might artificially introduce variability. To circumvent this effect, model output on different grids are remapped onto a common grid first by means of a conservative method (e.g. Kotlarski et al., 2014; Katragkou et al., 2015). Although the ECMWF forecast model is based on a discretisation scheme that turns the governing equations into truncated spherical harmonics, the area assumption also holds for the forecast products (http://www.ecmwf.int/sites/default/files/elibrary/2015/16559-user-guide-ecmwf-forecast-products.pdf - page 15).

We are well aware of these approaches as early as we started to evaluate the huge D-PHASE multi-model ensemble provided for the COPS campaign in 2007 (Bauer et al.,2011) and as members of CORDEX-Europe since 2010. The model comparison strategies were subject of extensive discussion in the D-PHASE Steering Committee of WWRP. Please note that also remapping on a common grid has its limitations, as pointed out by the WWRP JWGFVR among others and also driven by our own previous research, as this procedure causes interpolation errors for each model. However, we agree that there is no way around to proceed accordingly to realize reasonable comparisons but the results should just be interpreted with care due to the interpolation errors.

Bauer, H.-S., Weusthoff, T., Dorninger, M., Wulfmeyer, V., Schwitalla, T., Gorgas, T., Arpagaus, M. and Warrach-Sagi, K. (2011), Predictive skill of a subset of models participating in D-PHASE in the COPS region. Q.J.R. Meteorol. Soc., 137: 287–305. doi:10.1002/qj.715

Katragkou, E., M. Garcia-Diez, R. Vautard, S. Sobolowski, P. Zanis, G. Alexandri, R. M. Cardoso, A. Colette, J. Fernandez, A. Gobiet, K. Goergen, T. Karacostas, S. Knist, S. Mayer, P. M. M. Soares, I. Pytharoulis, I. Tegoulias, A. Tsikerdekis, and D. Jacob (2015), Regional climate hindcast simulations within EURO-CORDEX: evaluation of a WRF multi-physics ensemble, Geosci. Model Dev., 8(3), 603-618, doi: 10.5194/gmd-8-603-2015.

We followed your suggestion and tried conservative remapping using the ESMF regridding function as the native ARW grid is not supported e.g. by CDO. Unfortunately this procedure violates the NetCDF format constraints due to additional variables stored in the weighting file for this huge domain. This means that the regridding routines from ESMF and CDO have to be

rebuilt with NetCDF4 or even parallel NetCDF support which is currently out of reach. We are very happy to apply this idea in future studies, if a suitable infrastructure becomes available, and added the following paragraph on p. 6, line 12:

"We also tried conservative remapping using the ESMF regridding function as the native ARW grid is not supported by the Climate Data Operators (CDO, https://code.zmaw.de/projects/cdo). Unfortunately this procedure violates the NetCDF format constraints due to additional variables stored in the weighting file for this huge domain. This means that the regridding routines from ESMF and CDO have to be rebuilt with NetCDF4 support which is currently out of reach. According to a study of Jones et al. (1999), the conservative remapping method maintains the integral of the interpolated variable but may introduce larger interpolation errors which can introduce artificial variability."

Page 6, line 14: Kotlarski et al. (2014) did not evaluate "forecasts", but "hindcasts" or "evaluation runs".

Thank you for your comment. We replaced "forecasts" with "evaluation run".

Page 9, line 11: It is not clear how PSS is calculated: is it calculated from frequency distributions at each grid cell and then averaged over the sub-domains or is it based on frequency distributions from sub-domain averaged variables? A clarification of this would increase the traceability of the paper and could be added to the "Varification data strategy".

Thank you for your comment. The PSS is calculated at each grid cell and then averaged over the domain. This was clarified in the verification data strategy section on page 7, line 11:

"The PSS is calculated at each grid cell and finally averaged over the corresponding sub-domains in order to keep a large number of samples to calculate the PDFs as suggested by Perkins et al. (2007)."

Page 9, line 15: the simulations are driven by analysis fields. Due to the high internal variability, it cannot be concluded that there is a PSS when the model is forecast mode.

Perkins et al. (2007) applied their score to evaluate climate predictions. Therefore we are sure that there is a PSS even when the model is driven by forecast data.

Page 9, line 32-33: the high variability in HIRES may also be a consequence of the bilinear interpolation. When HIRES is conservatively remapped onto the ECMWF grid first, this source of variability can be avoided.

See our comment above.

Page 10, line 21: typo "... the the ..."

This typo was corrected.

Page 13, line 16: which forecasts have been used (initial time and steps)? Information on this would also fit into the "Verification data strategy".

We added a sentence on page 7, line 1:

"Additionally, the operational 24 hour forecast of the 00 UTC forecasts from ECMWF were applied to classify the WRF precipitation forecasts."

Page 14, line 26-27: Does this mean, in HIRES Kain-Fritsch was active together with the Morrison scheme? Or is there a typo?

Thank you for your comment. On the CP scale, the Kain-Fritsch scheme was not active. The sentence on page 14, starting line 33 was slightly changed to:

"As the strong negative pressure bias is not visible in the HIRES simulation, this indicates an unfavorable combination of the Kain-Fritsch convection parameterization with the Morrison microphysics scheme at the coarser resolution over the subpolar regions."

Page 15, line 8-13: The period from the July 26 to 28 was dominated by an extended trough in the eastern North Atlantic that developed from a large cyclone in the North Atlantic region from July 24 onwards. West and south to this cyclone a high pressure system developed (its centre lies approximately at 50°W/35°N). In total, this caused a drop of ~8.5 hPa during July 27 in the area averaged MSLP (see Figure 10). Means over large areas that average out these phenomena are not suitable to disentangle the boundary influences from the influences of SST and hence the sharp drop in MSLP as it is synchronously seen in all simulations as well as in the driving data cannot be convincingly be explained by the attempt on page 15 line 8-13, especially since Alexandru et al. (2007) demonstrated that internal variability depends on synoptic conditions independently from prescribed SST. A description of this issue can be easily used to demonstrate the difficulties that emerge when belt simulations are compared to observational based data (see General Comment).

We followed your suggestion and even further reduced the averaging domain to an area from 70°W-40°W and 30°N-45°N. The signal is still very similar to the results shown in Figure 10. The WRF model shows a too strong pressure drop compared to the ECMWF analysis.

Alexandru et al. (2007) applied a very coarse model grid in combination with only 18 vertical levels up to 30 km altitude which means roughly one layer every 2 km. Especially with the coarse resolution including convection parameterization it is questionable if the model is really capable to accurately simulate the spatial variability.

As the SST data are very close to each other in all three models, we assume that we can disentangle the boundary influence and the influence of SST data. As mentioned the center of

the pressure low is approx. at 50°W/35°N which is well inside the model domain and more than 1600km away from the boundary zone. As the zonal winds are also very weak during this period, we prefer to maintain our view in this case.

Concerning Kida et al. (1991): yes, they describe an alternative way to nest LAMs into coarser resolution models. My intention to refer to Kida et al. (1991) was that it is one of the earliest studies that drastically demonstrate the effect of internal variability (at that time it was not called "internal variability"). In addition, it laid down the principals of spectral nudging based on the ideas of Anthes et al. (1989). Nowadays, spectral nudging is implemented in a broad range of models, including WRF.
Anthes, R.A., Y.H. Kuo, E.Y. Hsie, S. Low-Nam and T.W. Bettge (1989), Estimation of skill and uncertainty in regional numerical models. Quart. J. Roy. Meteor. Soc., 115, 763-806.
For the same reason I referred to Peagle et al. (1996).

We are aware that spectral nudging is available for the WRF model system and that it is applied for downscaling purposes. However, as our intention is to use the model as a single realization of a seasonal forecast ensemble, we did not want to constrain the model to the "observation". In this case it is difficult to evaluate the applied model. There is also the risk that the model misses extreme events in case they are not present in the (coarser) model which serves as basis for spectral nudging.

We added a sentence on page 5, line 23 to further clarify our intention:

[revised manuscript text omitted]